# Improved transformation efficiency in *Mycoplasma hominis* enables disruption of the MIB–MIP system targeting human immunoglobulins

Jennifer Guiraud,[1,2] Chloé Le Roy,[1] Fabien Rideau,[3] Pascal Sirand-Pugnet,[3] Carole Lartigue,[3] Cécile Bébéar,[1,2] Yonathan Arfi,[3] Sabine Pereyre[1,2]

**ABSTRACT** The pathogenicity of *Mycoplasma hominis* is poorly understood, mainly due to the absence of efficient genetic tools. A polyethylene glycol-mediated transformation protocol was recently developed for the *M. hominis* reference strain M132 using the pMT85-Tet plasmid. The transformation efficiency remained low, hampering generation of a large mutant library. In this study, we improved transformation efficiency by designing *M. hominis*-specific pMT85 derivatives. Using the Gibson Assembly, the *Enterococcus*-derived *tet*(M) gene of the pMT85-Tet plasmid was replaced by that of a *M. hominis* clinical isolate. Next, the *Spiroplasma*-derived spiralin gene promoter driving *tet*(M) expression was substituted by one of three putative regulatory regions (RRs): the *M. hominis* arginine deiminase RR, the *M. hominis* elongation factor Tu RR, or the 68 bp SynMyco synthetic RR. SynMyco-based construction led to a 100-fold increase in transformation efficiency in *M. hominis* M132. This construct was also transformed into the *M. hominis* PG21 reference strain and three other clinical isolates. The transposon insertion locus was determined for 128 M132-transformants. The majority of the impacted coding sequences encoded lipoproteins and proteins involved in DNA repair or in gene transfer. One transposon integration site was in the mycoplasma immunoglobulin protease gene. Phenotypic characterization of the mutant showed complete disruption of the human antibody cleavage ability of the transformant. These results demonstrate that our *M. hominis*-optimized plasmid can be used to generate large random transposon insertion libraries, enabling future studies of the pathogenicity of *M. hominis*.

**IMPORTANCE** *Mycoplasma hominis* is an opportunistic human pathogen, whose physiopathology is poorly understood and for which genetic tools for transposition mutagenesis have been unavailable for years. A PEG-mediated transformation protocol was developed using the pMT85-Tet plasmid, but the transformation efficiency remained low. We designed a modified pMT85-Tet plasmid suitable for *M. hominis*. The use of a synthetic regulatory region upstream of the antibiotic resistance marker led to a 100-fold increase in the transformation efficiency. The generation and characterization of large transposon mutagenesis mutant libraries will provide insight into *M. hominis* pathogenesis. We selected a transformant in which the transposon was integrated in the locus encoding the immunoglobulin cleavage system MIB–MIP. Phenotypic characterization showed that the wild-type strain has a functional MIB–MIP system, whereas the mutant strain had lost the ability to cleave human immunoglobulins.

**KEYWORDS** *Mycoplasma hominis*, transposon mutagenesis, SynMyco, mycoplasma immunoglobulin protease

Address correspondence to Sabine Pereyre, sabine.pereyre@u-bordeaux.fr.

The authors declare no conflict of interest.

*M*ycoplasma hominis is a commensal bacterium and an opportunistic human pathogen associated with genital, neonatal, and extragenital infections (1, 2). Although the mechanism of transition from colonization to infection has not been established in *M. hominis*, candidate virulence factors include lipoproteins. For instance, the surface lipoprotein Vaa is a highly variable adhesin potentially involved in the first contact of *M. hominis* with its host cell (3, 4). OppA is a multifunctional protein essential for cell adhesion and ATPase-mediated damage (5). The lipoproteins P120 and P80 are recognized by the human humoral immune response (6–8). MHO_0730, a mycoplasma nuclease, counteracts neutrophil defenses by disrupting neutrophil extracellular traps (NETs) (9). However, many of the annotated coding DNA sequences of the *M. hominis* genome have unpredicted functions (10) and *M. hominis* pathogenicity is poorly understood, mainly due to the lack of efficient genetic tools to inactivate and complement putative virulence genes.

One of the current strategies used to edit mycoplasma genomes requires complex process consisting of genome cloning in yeast then back genome transplantation. The chromosome of a mycoplasma cell is transferred into a yeast cell, where it is processed as a very large plasmid. This bacterial chromosome can be edited using the large array of tools developed for yeast, and then transferred to a recipient bacterial cell, yielding a mutant mycoplasma cell. Indeed, this process was initiated in *M. hominis*, with the cloning of the *M. hominis* PG21 genome in yeast (11), but was hampered by the lack of a DNA transfer method.

*M. hominis* gene targeted mutants were generated using the targeting-induced local lesions in genomes (TILLING) strategy, allowing the characterization of two proteins involved in *M. hominis* pathogenicity (Vaa and OppA) (12). However, TILLING is a labor-intensive method and limiting the mutation frequency to a single mutation per genome is challenging.

Although transposon mutagenesis has been used in *Mycoplasma* species, in some species with high transformation rates (13–16), transformation assays in *M. hominis* have long yielded inconclusive results (4). A polyethylene glycol (PEG)-mediated transformation protocol using the pMT85-Tet plasmid was developed for the *M. hominis* M132 reference strain and two *M. hominis* clinical isolates (4, 17). However, the transformation efficiency remained low ($10^{-9}$ transformants/CFU/µg), hampering the generation of a large mutant library and investigation of the physiopathology of *M. hominis*. Several parameters may hamper transformation, including endonucleases of *M. hominis* (4), growth phase (4, 18), and the use of a plasmid not suited to *M. hominis*. Indeed, the pMT85-Tet plasmid harbors the *Spiroplasma*-derived spiralin gene promoter driving the *Enterococcus*-derived *tet*(M) gene expression (4, 17, 19).

In this study, the main goal was to improve the transformation efficiency in *M. hominis* by designing *M. hominis*-specific pMT85 derivatives. One of the constructs resulted in a marked increase in transformation efficiency, enabling the generation of a large mutant library. The transposon position in the transformants was determined, confirming insertion in nonessential loci. The second part of our study was devoted to the proof of concept that transposon mutagenesis allows gene disruption, and may thus enable the identification of putative virulence factors in *M. hominis*. To that extend, we analyzed the ability of a transformant with the transposon integrated in the MIP-encoding locus to cleave human immunoglobulins.

## MATERIALS AND METHODS

### Bacterial strains and culture conditions

The reference strains *M. hominis* M132 (ATCC 43521) and *M. hominis* PG21 (ATCC 23114) and the clinical isolates *M. hominis* 5012, *M. hominis* 4016, and *M. hominis* 4788 (20) were cultured at 37°C in Hayflick modified medium supplemented with arginine (HA) (21), for 18–24 h. Transformants were cultured in the same medium containing 2 µg/mL tetracycline. *Escherichia coli* NEB 5- competent cells (New England Biolabs) were used for

plasmid propagation and cloning. They were cultured at 37°C in Lysogeny Broth (LB) or LB agar. The tetracycline-resistant clinical isolates *M. hominis* 5860, *M. hominis* 6892, *M. hominis* 6227, *M. hominis* 5571, and *M. hominis* 6585 harboring the *tet*(M) resistance gene were from a prior study (22).

## *tet*(M) gene amplification and sequencing

PCR was performed on *M. hominis* DNA extracts obtained using the NucleoSpin Tissue Kit (Macherey-Nagel) from 200 µL culture, according to the manufacturer's instructions. A 1846 bp fragment of *tet*(M) was amplified from DNA extracts of the pMT85-Tet plasmid and the four *M. hominis* clinical isolates (5860, 6892, 6227, 5571, and 6585) using the primers tetM-D (5′-AAAACTACCTTAACAGAAAG-3′) and tetM-E (5′-CTTTATCTATCCGAC-TATTT-3′). Sequencing of PCR amplicons was performed using the following primers (Eurofins Genomics): tetM-D, tetM-E, tetM-C (5′-AAGATATGGCTCTAACAATT-3′), and tet2 (5′-CGAGATTCGGTTAGAGTATC-3′). Sequencing data were analyzed using BioEdit 7.2.5 software (Isis Pharmaceuticals, Inc.) and compared to the *tet*(M) sequences of the pMT85-Tet plasmid and the *M. hominis* strains Sprott (accession number CP011538.1) and *M. hominis* 2539 (NZ_CP026341.1).

## Plasmid construction

The pMT85-Tet plasmid (4, 17) served as a template for plasmid modification. First, the *Enterococcus*-derived *tet*(M) gene was replaced with that of *M. hominis* 6227. Next, the spiralin gene promoter driving *tet*(M) expression was replaced by one of three regulatory regions (RRs): the *M. hominis* arginine deiminase putative RR, the *M. hominis* elongation factor putative RR, and the SynMyco synthetic RR (16). *In silico* analysis, based on the genome sequence of the *M. hominis* PG21 reference strain (NC_013511.1) (10), was performed to design the putative RRs of the arginine deiminase and the elongation factor *tufA* genes (Fig. S1). The RRs were defined after screening for the main transcription and translation domains in the upstream intergenic regions −35 box, Pribnow box, and ribosomal binding site (RBS).

The first three plasmids were generated using the Gibson Assembly method (23) with NEBuilder HiFi DNA Assembly Master Mix (New England Biolabs). SynMyco-derived construction was performed using the Q5 Site-Directed Mutagenesis Kit (New England Biolabs). Details of plasmid construction are provided in Table S1. Modules of the pMT85-Tet plasmid and the generated plasmids and primer sequences are listed in Table S1.

According to the manufacturer's instructions, assembled products were transformed into competent *E. coli* NEB 5 cells. Transformants were plated on LB agar supplemented with 5 µg/mL tetracycline. Individual clones were picked and cultured in LB containing 5 µg/mL tetracycline, and the corresponding plasmid was isolated using NucleoSpin Plasmid EasyPure and NucleoBond Xtra Midi Kits (Macherey-Nagel). The constructs were verified by enzymatic digestion (*Bam*HI, *Hind*III, and *Pst*I) and Sanger sequencing (Eurofins Genomics). Before transformation into *M. hominis*, plasmids were methylated *in vitro* using the CpG methyltransferase from *Spiroplasma* spp. MQ1 (M.SssI, New England Biolabs) according to the manufacturer's recommendations.

## *Mycoplasma hominis* transformation protocol

*M. hominis* strains were transformed using a previously reported PEG-mediated transformation protocol with differences in pre-culture conditions (4). Briefly, a frozen −80°C stock culture of *M. hominis* at a known concentration was thawed and diluted to obtain $10^4$ colony forming units (CFU)/mL. The dilution was then inoculated in 10 mL of HA medium in sealed hemolysis tubes and incubated at 37°C without $CO_2$ until the culture reached the mid-logarithmic phase (24 h for the *M. hominis* PG21 reference strain, 13.5 h for *M. hominis* 4788, and 18 h for the *M. hominis* M132 reference strain and 5012 and 4016 isolates). Cells were harvested by centrifugation at 10,000× g for 20 min at

4°C and the pellet was washed twice with Tris Buffer. Cells were then suspended in $CaCl_2$ and incubated at 4°C for 30 min. $CaCl_2$-incubated cells (100 µL) were gently mixed with 10 µg of yeast tRNA (Life Technologies) and 10 µg of methylated plasmid. This mixture was poured onto 1.5 mL of 40% PEG 8000 (Sigma-Aldrich) for 30 min at room temperature then 7.5 mL of HA liquid medium were added before incubation for 3 h at 37°C. After centrifugation at 8,000× g for 10 min, the pellet was suspended in 1 mL of HA liquid medium and plated onto HA agar supplemented with 2 µg/mL tetracycline and incubated at 37°C in 5% $CO_2$. Bacterial titer was evaluated before transforming the cells by determining color changing units (CCUs) and CFUs/mL as described previously (4). The result of this assay was gained a few days after the transformation experiment and was used to calculate the transformation efficiency.

Transformation efficiency was calculated as the ratio of the number of CFUs on HA-tetracycline selective agar after transformation to the number of CFUs on HA agar before transformation per microgram of plasmid. The presence of *tet*(M) in the transformants was analyzed by PCR to confirm transposon insertion in the *M. hominis* genome, as reported previously (4). Transformants of interest were subcloned three times by successive passages on selective solid medium containing 2 µg/mL tetracycline.

## Determination of the transposon insertion site in *M. hominis* M132 transformants

Transposon insertion sites were determined by amplification and sequencing of the junction between the transposon and the adjacent genomic DNA, using single-primer PCR as described previously (with different primers) (4). The MT85–3 primer (5′-GTT-TCGCCACCTCTGACTTG-3′) was used for amplification and the nested primer MT85–2 (5′-TTACCGCCTTTGAGTGAGCT-3′) was used for Sanger sequencing (Eurofins Genomics). The sequences were compared to that of the *M. hominis* strain M132 genome for identification of the transposon insertion site, using the iterative Basic Local Alignment Search Tool (blastn) in the MolliGen 4.0 database (24) (http://www.molligen.org) or in the National Centre for Biotechnology Information (NCBI) nucleotide sequence database (http://www.blast.ncbi.nlm.nih.gov). Orthology assignments of the disrupted coding sequences (CDSs) were performed from protein sequences using COG mapper software (25) or the Basic Local Alignment Search Tool (blastp) in the NCBI database if there were no results in the MolliGen 4.0 database (24). Functional domain prediction was performed using InterproScan software in the Interpro database (26).

## MIB-MIP immunoglobulin cleavage assay

Remnant of serum from a female patient with a *M. hominis* infection was anonymously collected at the Bordeaux University Hospital, France. In this case, *M. hominis* was responsible for a post-partum endometritis. Serum was depleted of albumin using the Pierce Albumin Depletion Kit (Thermo Fisher Scientific) following the manufacturer's protocol to avoid distortion of the signal of immunoglobulins, which have a close molecular weight, in western blot experiments (27). Immunoglobulin cleavage assays were performed as described previously for *Mycoplasma mycoides* subsp. *capri* strain GM12 (27). Briefly, *M. hominis* M132 and *M. hominis* transformant 75–153 were inoculated into HA medium from frozen stock and cultured for 24 h at 37°C. Approximately $1 \times 10^8$ CFUs from cultures in the late exponential phase were harvested by centrifugation at $6,800 \times g$ for 10 min. The pellet was washed by resuspension in 1 mL fresh HA medium without foal serum ($HA_{\Delta FS}$) and centrifugation at $6,800 \times g$ for 10 min. The pellet was resuspended in 15 µL $HA_{\Delta FS}$ containing 2% (vol/vol) albumin-depleted human serum. Cleavage positive control reactions were established by adding 2 µg purified recombinant MIB (r-0583) and 2 µg purified recombinant MIP (r-0582) (27), to 15 µL $HA_{\Delta FS}$ containing 2% (vol/vol) albumin-depleted human serum. After 30 min of incubation at 37°C, the cells were pelleted by centrifugation at $6,800 \times g$ for 10 min. The supernatant was collected and mixed with 5 µL 4× Laemmli buffer containing β-mercaptoethanol and

denatured at 95°C for 10 min. Immunoglobulin heavy chain integrity was assessed by western blotting.

For western blotting, samples were separated by SDS-PAGE on a 10% acrylamide gel, and transferred to a Protran nitrocellulose membrane (Amersham) using a TE 77 Semi-Dry Transfer System (Amersham). Membranes were blocked overnight at 4°C in blocking buffer (phosphate-buffered saline with 0.05% vol/vol Tween 20 and 2% mass/vol bovine serum albumin), followed by incubation with the primary antibody diluted 1:2,000 in blocking buffer for 1 h at room temperature. Unbound primary antibody was removed by washing the membrane in wash buffer (PBS supplemented with 0.05% vol/vol Tween 20) three times for 10 min each at room temperature. Then the membrane was incubated with the HRP-conjugated secondary antibody diluted 1:2,000 in blocking buffer for 1 h at room temperature. Unbound secondary antibody was removed by washing the membrane in wash buffer (PBS with 0.05% vol/vol Tween 20) three times for 10 min each at room temperature. Detection was performed using the Pierce SuperWestPico Chemiluminescence Substrate (Thermo Scientific) and imaging was performed on a ChemiDoc MP imager (Bio-Rad). Human IgG or IgM heavy chain was detected using rabbit anti-human IgG-Fc and rabbit anti-human IgM (Bethyl) primary antibodies, and HRP-conjugated goat anti-rabbit IgG (Jackson ImmunoResearch) as the secondary antibody.

## RESULTS

### Transformation efficiencies of *M. hominis* M132

Efficient transformation may be related to better expression of the antibiotic resistance gene carried on the vector. Therefore, as a starting point for this study, we designed a modified pMT85-Tet plasmid bearing a tetracycline resistance marker that would be more adapted to *M. hominis*. We hypothesized that a *tet*(M) gene variant from a clinical strain might have higher expression than its heterologous counterpart from *Enterococcus faecalis*. Indeed, the *Enterococcus*-derived *tet*(M) gene carried by the original plasmid exhibited several single nucleotide polymorphisms compared to the *tet*(M) genes in the genomes of seven *M. hominis* isolates resistant to tetracycline (5860, 6892, 6227, 5571, 6585, Sprott, and 2539; Fig. S2). In addition, comparison of Tet(M) amino acid sequences revealed 13 amino acid substitutions, including eight shared by all of the *M. hominis* isolates (Fig. S2). Among the other five amino acid substitutions, three (M195L, T297I, and E345K) were found in a single *M. hominis* isolate, one (S557L) was found in two isolates, and one (S285L) was found in three *M. hominis* isolates. Given that the Tet(M) protein sequence of the *M. hominis* 6227 harbored amino acid changes shared by most *M. hominis* strains, we considered this sequence as a consensus and selected it for further development. Despite four independent transformation assays, no transformants were obtained using the construct harboring *tet*(M) from a clinical strain (Fig. 1). Thus, we replaced the spiralin gene promoter driving *tet*(M) expression with one of three RRs. We focused on essential genes with high transcription rates. We selected the putative RR of the *M. hominis* arginine deiminase gene, an enzyme involved in arginine hydrolysis and energy metabolism (10). *In silico* analysis of the 448 bp region between the arginine deiminase gene and the adjacent gene encoding a hypothetical protein located on the opposite-sense strand revealed that this 448 bp intergenic region likely contained the RRs of both genes (Fig. S1). Consequently, the 212 bp region upstream of the arginine deiminase gene, which likely contained the transcriptional and translational main domains of only the arginine deiminase gene, was selected as an RR for driving *tet*(M) expression (Fig. S1). Despite three independent transformation assays, no transformants were obtained using the plasmid carrying the *M. hominis* arginine deiminase RR (Fig. 1). Next, we selected the entire intergenic region upstream of the elongation factor Tu gene (Fig. S1), a highly conserved protein involved in protein biosynthesis, and the 68 bp synthetic sequence of the SynMyco RR designed from the RRs of the elongation factor Tu orthologues in a set of *Mycoplasma* species (16). The *M. hominis* reference strain M132 was successfully transformed at least three times in six independent transformation

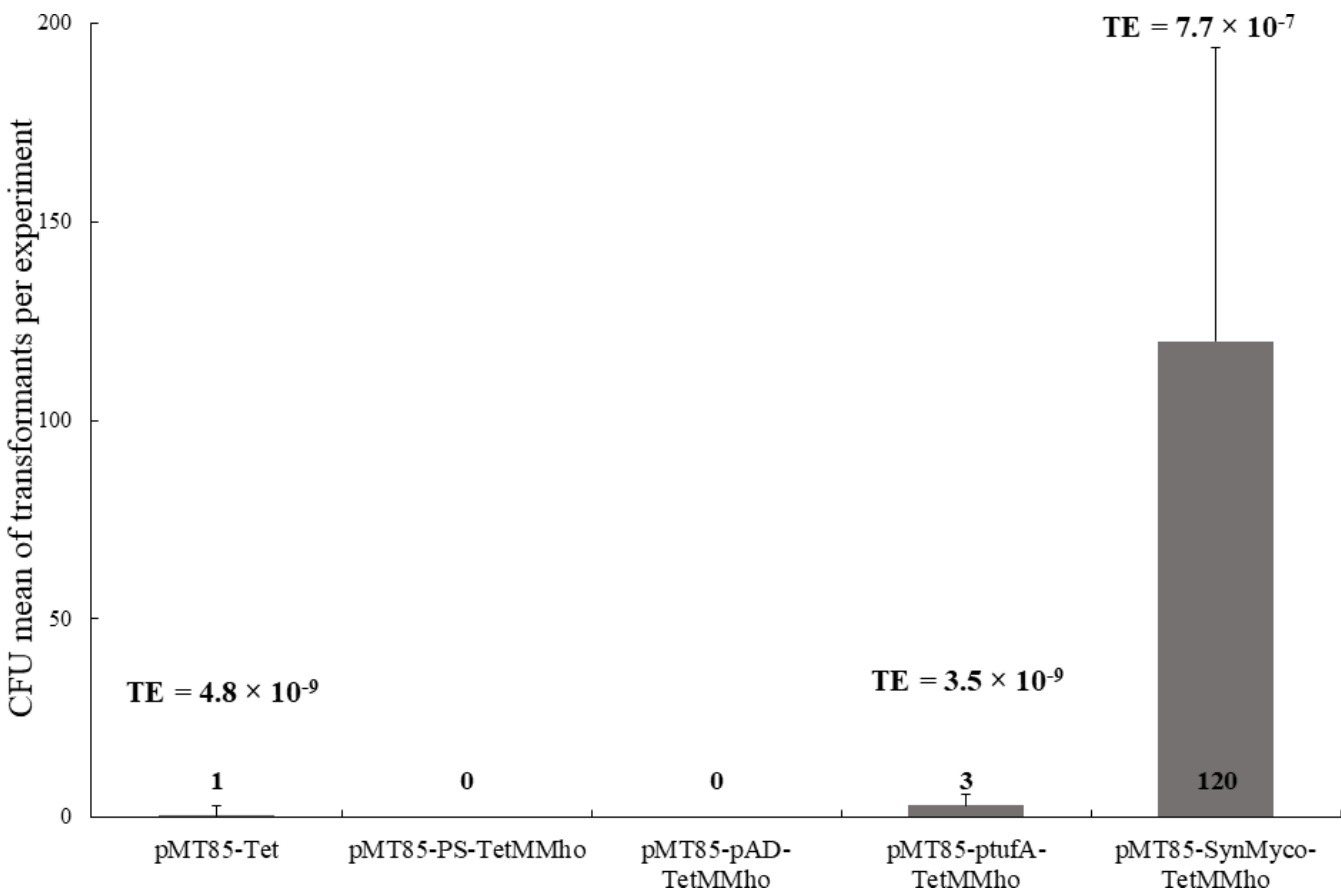

**FIG 1** Numbers of *M. hominis* transformants per experiment and transformation efficiencies for *M. hominis* M132 using the original plasmid pMT85-Tet or the four pMT85-derived constructs. Bars show median numbers of transformants in independent replicates with each construct; error bars are standard deviations. Average transformation efficiencies (TE), (transformants/CFU/µg plasmid) are indicated above bars.

experiments with the original pMT85-Tet vector (three of six experiments), the construct harboring the *tuf*A RR (six of six experiments), or with the SynMyco-construction (six of six experiments) (Fig. 1; Table 1).

The original pMT85-Tet plasmid had a transformation efficiency of $4.8 \times 10^{-9}$ transformants/CFU/µg plasmid, corresponding to zero to six transformants per experiment. The transposon vector carrying the *tufA* RR did not increase the transformation efficiency ($3.5 \times 10^{-9}$ transformants/CFU/µg; two to nine transformants per experiment). However, the transformation efficiency was increased 100-fold using the SynMyco-construction ($7.7 \times 10^{-7}$ transformants/CFU/µg; 120 transformants per experiment). A total of 806 transformants was obtained for *M. hominis* M132 using pMT85-SynMyco-tetMMho in six experiments. Transformed *M. hominis* M132 colonies typically appeared on selective agar after 5–7 days of incubation, and had a typical morphology. Picked colonies grew in selective liquid medium after 24–48 h of incubation but up to 10 days in some cases.

### Transformation of four *M. hominis* reference or clinical strains

The transformation protocol using the SynMyco construct was tested using the *M. hominis* PG21 reference strain and the 4788, 4016, and 5012 *M. hominis* clinical isolates (Table 1). Because these strains have different growth rates, we modified the incubation time to ensure that mid-logarithmic phase cultures were used for transformation. All of the tested strains were transformed; the highest efficiency was for the *M. hominis* reference strain PG21 ($4.7 \times 10^{-7}$ transformants/CFU/µg, 71 PG21 transformants in

**TABLE 1** Transformation experiments performed in this study

| Transformed *M. hominis* strain | Plasmid name | RR[a]-version | *Tet*(M) origin | No. of successful to total transformations | Mean number of transformants per experiment | Transformation efficiency | Significance[b] (*t*-test) |
|---|---|---|---|---|---|---|---|
| M132 | pMT85-Tet | Spiralin promoter | *E. faecalis* | 3/6 | 1 | $4.8 \times 10^{-9}$ | |
| M132 | pMT85-PS-tetMMho | Spiralin promoter | *M. hominis* | 0/4 | 0 | - | |
| M132 | pMT85-pAD-tetMMho | *M. hominis* arginine deiminase gene RR | *M. hominis* | 0/3 | 0 | - | |
| M132 | pMT85-ptufA-tetMMho | *M. hominis* elongation factor gene RR | *M. hominis* | 6/6 | 3 | $3.5 \times 10^{-9}$ | $P = 0.39$ |
| M132 | pMT85-SynMyco-tetMMho | SynMyco RR | *M. hominis* | 6/6 | 120 | $7.7 \times 10^{-7}$ | $P = 0.10$ |
| PG21 | pMT85-SynMyco-tetMMho | SynMyco RR | *M. hominis* | 2/2 | 36 | $4.7 \times 10^{-7}$ | |
| 4788 | pMT85-SynMyco-tetMMho | SynMyco RR | *M. hominis* | 1/1 | 2 | $1.3 \times 10^{-8}$ | |
| 5012 | pMT85-SynMyco-tetMMho | SynMyco RR | *M. hominis* | 1/1 | 4 | $4.2 \times 10^{-9}$ | |
| 4016 | pMT85-SynMyco-tetMMho | SynMyco RR | *M. hominis* | 1/1 | 6 | $2.2 \times 10^{-9}$ | |

[a]RR; regulatory region.
[b]One-tailed *t*-test *P*-values are indicated for transformation efficiency obtained with the pMT85-ptufA-tetMMho and the pMT85-SynMyco-tetMMho plasmids compared to the transformation efficiency obtained with pMT85-Tet plasmid.

two experiments). Two transformants were obtained for the 4788 isolate ($1.3 \times 10^{-8}$ transformants/CFU/µg). The transformation efficiencies of isolates 5012 (four transformants) and 4016 (six transformants) were $4.2 \times 10^{-9}$ and $2.2 \times 10^{-9}$ transformants/CFU/µg, respectively.

## Transposon insertion sites

The transposon insertion position was determined via single-primer PCR followed by Sanger sequencing of the junction between the transposon and the mycoplasma genome. Single-primer PCR was performed on 527 *M. hominis* M132 transformants obtained in three transformation assays using pMT85-Tet, pMT85-ptufA-tetMMho, or pMT85-SynMyco-tetMMho. Successful results were obtained for 128 (24%) transformants. For the other transformants, the insertion sites could not be precisely determined because of uninterpretable chromatograms, mainly due to several superimposed peaks in the sequencing profile, suggesting insertion of several copies of the transposon.

The transposon was inserted in a coding region in 108 of the 128 transformants (84.4%); for 20 transformants (15.6%), the transposon was inserted in an intergenic region (Fig. 2; Table S2). Overall, a total of 65 DNA coding sequences harbored a transposon (Fig. 2). Although the insertion events were likely randomly distributed in the M132 genome, some regions appeared to have been spared, notably regions coding for tRNAs, ribosomal proteins, DNA polymerase subunits, and enzymes involved in energy production or nucleotide metabolism. Conversely, the frequency of insertion was higher in a small number of CDSs. For instance, 17 events were identified in the predicted F1-like$X_0$ ATPase subunit alpha homolog (Mhom132_03120) and 14 events were identified in the hypothetical protein MHO_1170 ($n = 14$) (Table S2). Overall, most of the CDSs in which insertion occurred encoded putative lipoproteins ($n = 22$) such as Lmp proteins, membrane nucleases, or ABC transporters; and proteins involved in the DNA repair machinery ($n = 7$) or in intracellular trafficking, secretion, and vesicular transport ($n = 7$). In addition, 13 *M. hominis* M132 transformants had transposon insertions in the *M. hominis* integrative and conjugative element (ICE).

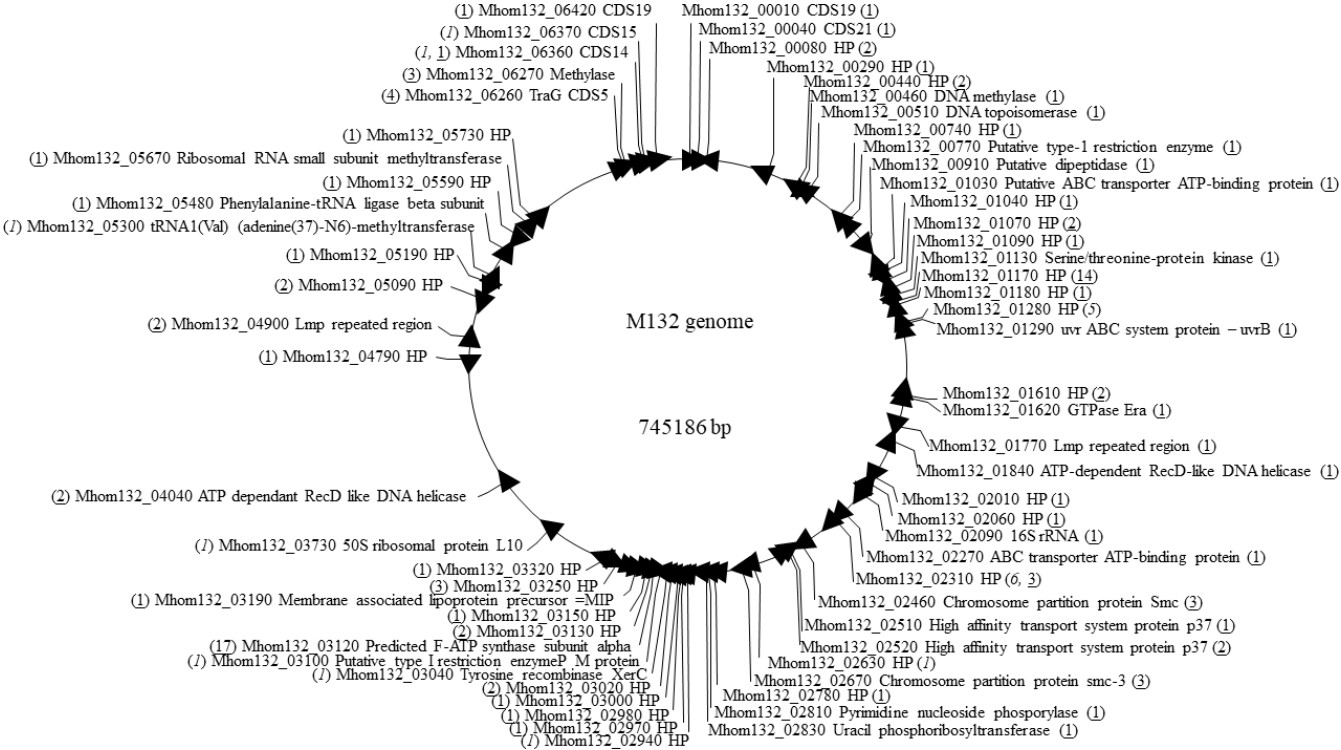

**FIG 2** Transposon positions in the *M. hominis* M132 genome in the 128 characterized transformants. Arrowheads indicate CDSs targeted for transposon insertion. M132-related mnemonics and corresponding CDS annotations are based on The MolliGen database. The number of M132-transformants per locus is shown in parentheses. Underlined and italicized numbers indicate transposons inserted in coding and intergenic regions, respectively. In the latter case, the CDS located immediately downstream of the transposon is indicated. HP, hypothetical protein.

## Transposon integration in the Mhom132_03190 locus disrupts the MIB-MIP system

In *M. hominis* 75–153, the transposon was inserted in the Mhom132_03190 locus, encoding a putative MIP (Fig. 3; Table S2). This protease with the Ig-binding protein MIB is part of a well-characterized mycoplasma system that degrades antibodies (28). The MIB–MIP homolog system in *M. hominis* M132 is encoded by two genes, Mhom132_03190 (MIP) and Mhom132_03180 (MIB) (Fig. 3A). Given that the transposon is inserted at the beginning of the Mhom132_03190 locus at position 179 of 2427 (Fig. 3B), we hypothesized that transposon integration disrupted the system, which is putatively transcribed as a single operon (29). To test this hypothesis, we assessed the abilities of the *M. hominis* M132 wild-type and 75–153 transformant to cleave antibodies (Fig. 3C and D). After incubation with human serum, the integrity of immunoglobulin G and M heavy chains was evaluated by western blotting. As the positive control, purified recombinant MIB and MIP proteins were added to serum, which yielded 44 and 66 kDa fragments indicative of cleavage of the heavy chains of IgG and IgM. Wild-type *M. hominis* M132 cleaved immunoglobulins G and M, indicating a functional MIB–MIP system. This proteolytic activity was abolished in the 75–153 transformant, because only intact immunoglobulin G and M heavy chains were detected, confirming that transposon integration in the Mhom132_03190 locus led to protein disruption and loss of MIB–MIP activity.

## DISCUSSION

Much effort has focused on developing an optimal PEG-mediated transformation protocol for *M. hominis* M132 using pMT85-Tet. Several attempts to improve transposon mutagenesis resulted in the first generation of *M. hominis* transformants (4). However,

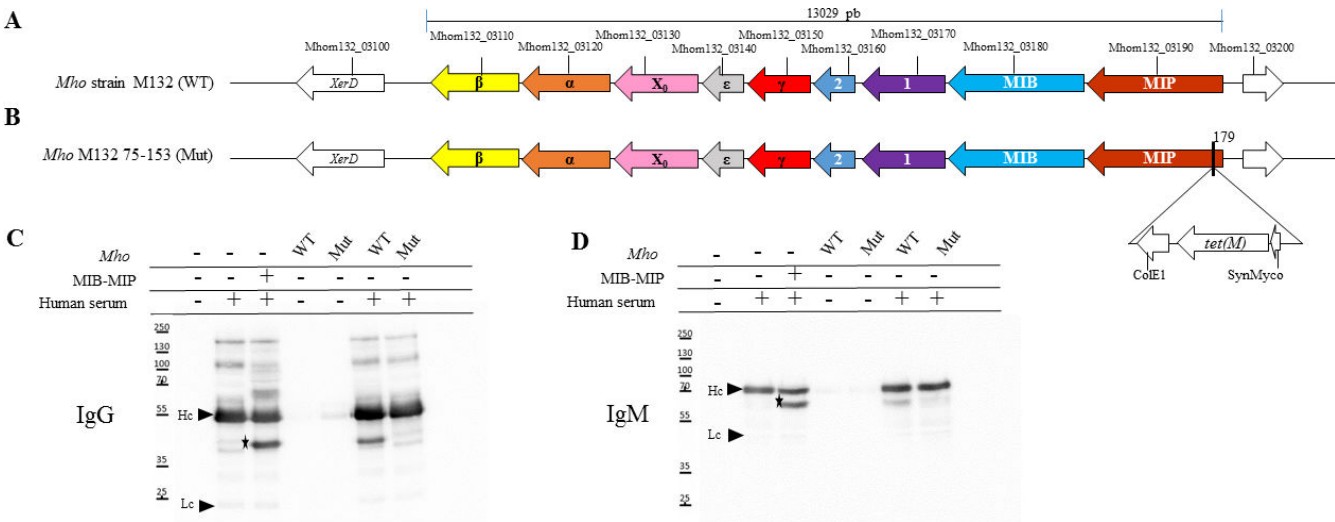

**FIG 3** Active MIB–MIP system in the *M. hominis* wild-type strain M132 and disrupted in the 75–153 *M. hominis* transformant. (A) Schematic of orthologous CDSs in the *M. hominis* (*Mho*) wild-type strain M132 encoding MIB (blue), MIP (red), and the subunits of the putative F1-like$X_0$ ATPase (β yellow, α orange, $X_0$ pink, ε gray, γ red, and putative proteins 1 purple and 2 blue). CDSs are indicated with their corresponding mnemonic in the M132 genome; the color code is from the first functional description (28, 29). (B) Genomic loci encoding the MIB–MIP system in the 75–153 *M. hominis* transformant with the transposon inserted at position 179 of 2,427 bp of the Mhom132_03190 locus. (C) and (D) Immunoglobulin cleavage by recombinant MIB and MIP proteins, wild-type (WT) *M. hominis*, and the 75–153 *M. hominis* transformant (Mut). Assays were performed using serum from patient infected by *M. hominis*. Samples were split in two and analyzed by western blotting against human IgG (C) and IgM (D). Arrowheads indicate intact immunoglobulin heavy chains (Hc) and light chains (Lc). Asterisks indicate cleaved immunoglobulin heavy chain.

the transformation efficiency remained low. In this study, we improved transformation efficiency by designing an *M. hominis*-specific pMT85 derivative.

We first replaced the *Enterococcus*-derived *tet*(M) gene with one from an *M. hominis* clinical isolate to enhance its expression in *M. hominis*. Although five amino acid substitutions linked to charge or hydrophobicity differed between the *M. hominis* Tet(M) sequence and that of pMT85-Tet, no transformants were obtained in three experiments. Indeed, the prediction of the possible impact of these five amino acid substitutions (namely H214Q, P251Q, N257K, K265E, and S285L) using PolyPhen-2 revealed a probable minor impact on the protein structure, explaining the absence of clear difference in transformation efficiency observed between both conditions ($10^{-9}$ vs $<10^{-9}$). We hypothesized that the low transformation efficiency might be related to the poor recognition of the spiralin promoter by *M. hominis*. Three putative RRs were tested to drive the *tet*(M) gene expression, including those of genes with high transcription rates in *M. hominis* [arginine deiminase (10) and elongation factor Tu (16, 30)]. Indeed, arginine deiminase mediates arginine hydrolysis, the main ATP-generating pathway in *M. hominis* (10). However, the chosen 212 bp putative regulatory region including the main sequence determinants (Fig. S1) was not suitable for efficient *M. hominis* transformation (16, 31). Elongation factor Tu is involved in translation and cell adhesion in prokaryotes (16, 30, 32). Plasmids harboring the *tufA* gene promoter have been used to transform other *Mollicutes* species (33, 34). Considering that an A/T-rich block likely having a stimulatory function has been previously identified upstream of the *tufA* promoter, we selected the entire intergenic region upstream of the elongation factor Tu gene (35). However, in our study, the transformation efficiency was not significantly improved by replacing the spiralin promoter with the entire intergenic sequence preceding the *tufA* gene, although transformation reproducibility was enhanced. Finally, we used the SynMyco synthetic RR, which is a consensus sequence from the *tufA* RRs of 10 *Mycoplasma* species (16). Its sequence was reduced to the five main domains to enhance recognition and the expression levels of resistance markers in a broad

range of mycoplasmas. This final iteration of our design yielded a 100-fold increase in transformation efficiency, enabling the generation of more than 800 transformants in five *M. hominis* strains. Similar results have been reported by Montero-Blay et al. for *Mycoplasma gallisepticum*, with the highest fold-change (1,000-fold) in transformation efficiency being for *Mycoplasma feriruminatoris* and *Mycoplasma agalactiae* (16). Our results thus confirm that the RRs have marked effects on transformation efficiency (16) and that the SynMyco-based construct is suitable for *M. hominis*. These results suggest that use of a shorter sequence for the RRs of the arginine deiminase and *tufA* genes could further increase the transformation efficiency (16, 33, 34), (Fig. S1D). In addition, it would be interesting to evaluate the benefit of replacing the tetracycline resistance gene with another antibiotic resistance marker commonly used in mycoplasmas, such as gentamicin and puromycin resistance genes (13–16, 18, 36, 37). Alternatively, the native RR of the antibiotic resistance marker could be added to the plasmid. For example, the impact of the 6227 *tet*(M) gene RR on the transformation efficiency could be evaluated in further attempts.

*M. hominis* transformants were obtained using only midlogarithmic phase cultures. Despite similar procedures as for *M. pneumoniae* and *M. arthritidis* (37, 38), efficiency was highest for *Ureaplasma parvum*, *M. agalactiae*, and *M. bovis* using bacterial cultures in the late exponential phase or a mixture of early-, middle-, and late-logarithmic phases (4, 13, 16, 18, 39). The growth phase at the time of transformation is related to transformation efficiency in mycoplasmas. In addition, transformation rates were higher with reference strains than clinical isolates, possibly because of the shorter exponential growth phase of the clinical isolates (data not shown). There were no differences in transformation efficiency for the *M. hominis* clinical isolates 4016 and 5012 transformed with the SynMyco-based construct or the pMT85-Tet plasmid (4). Consequently, to further enhance transformation efficiency, the transformation protocol for clinical isolates needs to be optimized and/or DNA defense mechanisms that hamper transformation need to be identified (36, 40, 41). Although few prior studies have focused on mycoplasmas (42, 43), we previously reported that diverse restriction-modification systems were present in multiple copies in *M. hominis* isolates, potentially hampering their uptake of foreign DNA (4).

In the present study, we performed single-primer PCR to identify transposon insertion sites. Successful results were obtained for 128 M132-transformants, but the yield of 24% is insufficient for larger studies of *M. hominis*. High-throughput transposon sequencing methods have been developed based on linear PCR enrichment of amplicons from the transposon-junction sequence followed by next-generation sequencing (14, 16, 44–46). Although sequencing costs have decreased, high-throughput methods and bioinformatics analysis are expensive and time consuming.

Analysis of the transposon insertion sites of the 128 *M. hominis* M132 transformants yielded a total of 65 CDSs, including 9 of unknown function and no homolog. Although the transposons were randomly inserted in the *M. hominis* genome, further transformation experiments are needed to identify transposon-insertion hotspots. Of the transformants, 84.4% integrated the transposon in a coding region, suggesting inactivation of the gene. Indeed, given the functional domains predicted in the CDSs (24–26) and that a gene is considered inactivated if the transposon insertion results in truncation of at least 10% of the 5'-end or 15% of the 3'-end of a CDS (14), transposon insertions likely affected protein functionality in >89.8% (97 of 108) of cases. In intergenic regions, transposons were inserted, with rare exceptions, in the 100-bp region upstream of the ATG start codon of the ORF, which likely harbors the promoter. Transposon insertions in the RR might thus contribute to the non-expression of adjacent genes. Overall, the majority of the impacted CDSs encoded lipoproteins and proteins involved in DNA repair or in gene transfer. However, further transformation experiments and characterization of insertion sites are needed to determine whether core gene regions may be less affected than accessory gene regions. Interestingly, 13 of *M. hominis* M132 transformants integrated the transposon in the ICE. Although ICEs are widely

distributed in *Mycoplasma* species and are implicated in horizontal gene transfer, their functions and transfer capacities in *M. hominis* are unclear (20, 47, 48). The generation of transformants carrying an ICE tagged with an antibiotic resistance gene marker would enable evaluation of the functionality of the ICE in mating experiments (49, 50). Our improved transposon mutagenesis method will enable the characterization of candidate virulence factors of *M. hominis*. The genetic variability of the lipoproteins P120, P80, and Lmp (6, 7, 51, 52); the immunogenicity of lipoproteins P120 and P80 (6, 8); and the role of lipoprotein p37 in cell invasiveness (53) are under investigation. Because several *M. hominis* transformants integrated the transposon in the genes encoding these surface lipoproteins, their phenotypic characterization may provide insight into the role of these lipoproteins in the pathogenesis of *M. hominis*. Subsequently, further development of *ori*C replicative plasmids adapted to *M. hominis* and retaining the SynMyco RR would facilitate complementation of mutants (54–56). Finally, the development of a further targeted mutagenesis method would be even more efficient to detect and characterize virulence genes (54–56). Development of a CRISPR-Cas base editor system, as already available for other *Mycoplasma* species, would also enable functional studies in *M. hominis* (57). Whole-genome transplantation assays could be conducted with *M. hominis* M132 as recipient cells and SynMyco-tetMMho-marked genomes from *M. hominis* PG21 extracted from yeast.

Among the 128 characterized transformants, the 75–153 transformant integrated the transposon in the 5′-end region of the *M. hominis* MIP-encoding CDS. This protease with MIB makes up the MIB–MIP system, which binds and cleaves immunoglobulins (27, 28, 58). This system, which is putatively involved in immune evasion by mollicutes, was first functionally characterized in *M. mycoides* subspecies *capri* (28). In this study, the MIB–MIP homolog system in the *M. hominis* M132 wild-type was functional, as indicated by cleavage of the heavy chains of immunoglobulins G and M. Phenotypic characterization of the *M. hominis* 75–153 transformant showed disruption of human antibody cleavage. In addition, as shown in Fig. 3, MIB and MIP genes are followed by seven homolog CDSs encoding the subunits of the predicted F1-likeX$_0$ ATPase, which is likely involved in immunoglobulin cleavage (29). A total of 20 *M. hominis* transformants integrated the transposon in three of the seven CDSs encoding the subunits of the predicted F1-likeX$_0$ ATPase, Mhom132_03120 (17 transformants with five distinct transposon insertion sites), Mhom132_03130 (two transformants with the same insertion site), and Mhom132_03150 (one transformant) (Table S2). Phenotypic characterization of these transformants will enable assessment of the involvement of these subunits in the MIB–MIP system.

In conclusion, the RR of the antibiotic resistance marker gene is important for improving transformation efficiency in *M. hominis*. The 68 bp-SynMyco synthetic RR will enable the generation of random transposon insertion libraries in *M. hominis*, in turn providing insight into the physiopathology of this human urogenital species.

## ACKNOWLEDGMENTS

We thank Quentin Jehanne and Philippe Lehours for assisting with the bioinformatics analysis.

This study received no specific grant from a funding agency in the public, commercial, or not-for-profit sector.

The study was conceptualized and supervised by F.R., P.S.P., C.L., C.B., Y.A., and S.P., and the investigation was conducted by J.G., C.L.R., P.S.P., and Y.A. The original manuscript draft was written by J.G., Y.A., and S.P. C.L.R, F.R., P.S.P., C.L., and C.B. reviewed and edited the manuscript.

The authors declare that they have no conflicts of interest to report.

## AUTHOR AFFILIATIONS

[1]Centre national de la recherche scientifique (CNRS), UMR 5234 Fundamental Microbiology and Pathogenicity, University of Bordeaux, Bordeaux, France
[2]Bacteriology Department, National Reference Centre for Bacterial Sexually Transmitted Infections, Bordeaux University Hospital, Bordeaux, France
[3]INRAE, BFP, UMR 1332, Univ. Bordeaux, Villenave d Ornon, France

## AUTHOR ORCIDs

Fabien Rideau  http://orcid.org/0000-0003-3626-5986
Pascal Sirand-Pugnet  http://orcid.org/0000-0003-2613-0762
Carole Lartigue  http://orcid.org/0000-0001-5550-7579
Yonathan Arfi  http://orcid.org/0000-0002-6064-7899
Sabine Pereyre  http://orcid.org/0000-0003-1757-4206

## AUTHOR CONTRIBUTIONS

Jennifer Guiraud, Data curation, Investigation, Writing – original draft | Chloé Le Roy, Data curation, Investigation, Writing – review and editing | Fabien Rideau, Conceptualization, Supervision, Writing – review and editing | Pascal Sirand-Pugnet, Conceptualization, Investigation, Supervision, Writing – review and editing | Carole Lartigue, Conceptualization, Supervision, Writing – review and editing | Cécile Bébéar, Conceptualization, Supervision, Writing – review and editing | Yonathan Arfi, Conceptualization, Investigation, Supervision, Writing – original draft | Sabine Pereyre, Conceptualization, Supervision, Writing – original draft

## DATA AVAILABILITY

The genome sequence of the *M. hominis* strain M132 is available in the NCBI database under accession number JASCXH000000000 in BioProject PRJNA493181.

## ETHICS APPROVAL

Informed consent is not required in France for the use of remnants of microbiological samples in the quality assurance of diagnostic methods according to the national legislation. The collection and use of anonymized samples were approved by the French Personal Data Protection Authority (CNIL, n°10.362).

## ADDITIONAL FILES

The following material is available online.

### Supplemental Material

**Figure S1 (Spectrum01873-23-s0001.docx).** *In silico* analysis of the putative regulatory regions.
**Figure S2 (Spectrum01873-23-s0002.docx).** Multiple sequence alignment of tetracycline resistance variants.
**Table S1 (Spectrum01873-23-s0003.xlsx).** pMT85-Tet derived plasmids and primers.
**Table S2 (Spectrum01873-23-s0004.xlsx).** Transposon insertion locations in the *M. hominis* 132 genome of the 128 characterized transformants.

### Open Peer Review

**PEER REVIEW HISTORY (review-history.pdf).** An accounting of the reviewer comments and feedback.

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
