## [Reviewer comments · Microbiology Spectrum]

Microbiology Spectrum

Improved transformation efficiency in *Mycoplasma hominis* enables disruption of the MIB-MIP system targeting human immunoglobulins

Jennifer Guiraud, Chloé Le Roy, Fabien Rideau, Pascal Sirand-Pugnet, Carole Lartigue, Cécile Bébéar, Yonathan Arfi, and Sabine Pereyre

Corresponding Author(s): Sabine Pereyre, Univ. Bordeaux, CNRS

Review Timeline:

Submission Date:	May 5, 2023
Editorial Decision:	May 30, 2023
Revision Received:	July 6, 2023
Editorial Decision:	July 11, 2023
Revision Received:	July 17, 2023
Accepted:	July 18, 2023

Editor: Olivier Neyrolles

Reviewer(s): The reviewers have opted to remain anonymous.

Transaction Report:

DOI: <https://doi.org/10.1128/spectrum.01873-23>

May 30, 2023

Prof. Sabine Pereyre
Univ. Bordeaux, CNRS
UMR 5234 Fundamental Microbiology and Pathogenicity
146 rue Léo Saignat
Bordeaux 33076
France

Re: Spectrum01873-23 (Improved transformation efficiency in *Mycoplasma hominis* enables disruption of the MIB-MIP system targeting human immunoglobulins)

Dear Prof. Sabine Pereyre,

Thank you for submitting your manuscript to Microbiology Spectrum.

Link Not Available

Sincerely,

Olivier Neyrolles

Journals Department
Reviewer comments:

Reviewer #1 (Comments for the Author):

The manuscript of Guiraud et al. deals with the description of an improved transformation efficiency of *Mycoplasma hominis* using a genetically modified tetM-plasmid and the characterization of a respective disruption of the MIB-MIP system targeting human immunoglobulins.

There are these two main topics, genetics for an improved transformation efficiency and characterization of the MIB-MIP system disruption, whose combination led to the question, what the goal of the study was.

The following comments may help to make the manuscript more complete on one side or the other.

Major comments

1. For better understanding of the transformation part, a table should be added comprising ALL transformation experiments described in the results with columns of 1. transformed *M. hominis* strain/ 2. plasmid-name/ 3. RR-version / 4. tetM-version/origin / 5. number of successful to total transformations / 6. number of transformants per experiment / 7. transformation efficiency.

Is it possible to calculate significance for transformation efficiency per plasmid ?

2. Why do the authors focus on strain M132 (as reference) and not PG21 (as type strain)? It is very difficult to follow and comprehend the transposon insertion sites without having access to an ncbi-deposited genome sequence with annotated genes and proteins with postulated functions. The BioSample accession number SAMN33331381 and BioProject PRJNA493181 do neither contain the genome sequence nor genes or proteins, and using Molligen I was unable to find the genes and encoded proteins described in a simple way. The same transformation analysis was done with PG21, a well-characterized type strain on genetic and protein level with the further advantage of carrying less mobile genetic elements and nearly representing a core-genome.

Thus, without having a comfortable way to get information about the affected M132 genes, it would be necessary to insert a column to the table of transposon insertion sites with accession numbers of the homologs and, in case of PG21 homologs, the MHO_XXXX numbers in addition.

3. In Figure S1 the reverse complementary strand is given but also ending with ATG. Is it correct and represents the start codon of an upstream positioned gene on that strand? However, it is not correct to entitle the reverse complementary strand with the sequence orientation of 3'-5'. It is always 5'-3' and should be entitled as "reverse complementary strand".

4. Multiple sequence alignments of RR- and tetM-variants should replace the display of individual sequences of the same region/ gene part in different species. Differences in *M. hominis* and *Enterococcus* tetM should be shown (line 220 and Fig.S2/3).

If homology of RR regions is too low, a multiple scheme alignment of RR-composition (sequence of (TGC)-Xn-(TAWAAT)-Xn-.... - (TGC)-Xn-(TAWAAT)-Xn - (ATG)) per species/variant would be informative and even here the acc-number of the sequences should be given with the information about the part of the region shown.

Is it known, whether sequences further upstream than the ATG-proximal (TGC)-Xn (TAWAAT) motifs are involved in gene regulation? Why were the full intergenic sequences displayed?

5. Figures S2 and S3 could be fused with nucleotide sequences given as consensus with wobble IUPAC-IUB symbols and the consensus amino acid sequence above with an indication of isofunctional amino acid changes. That would be sufficient to see variabilities in gene and protein sequences.

Non-isofunctional amino acid changes should be discussed in the text, as their impact on the protein structure will be higher than that of the isofunctional ones.

6. What was the rationale to characterise a more rare insertion event? Was it done as "proof-of-principle"? Only three times the MIB-MIP system was affected, of which only one was further characterized. The demonstration of MIB-MIP-dependent loss of human immunoglobulin cleavage is good, but is this sufficient to demonstrate the utility of an improved, but still random, transposon insertion as a tool for detecting and characterizing virulence genes?

However, it looks like as if transposon integration dominantly takes place in hotspots of variant lipoproteins, MGEs, defence islands etc. and will thus have the potential to rather destroy virulence-associated structures than core genes. Would it strengthen the line of the manuscript in mapping the transposon insertion sites to the *M. hominis* (PG21) genome and to analyse whether core gene regions are less affected than more variant (hotspot-like) regions? Would it be possible to calculate significances between both groups?

7. Minor comments

Line 61f: What does the authors want to express with "...colonization to invasiveness is not established...?"

Invasiveness of *M. hominis* in cells is rarely, but well documented (e.g. Taylor-Robinson et al., International journal of experimental pathology. 1991; Gdoura et al., BMC infectious diseases. 2007; Diaz-Garcia et al., Hum Reprod. 2006; Hopfe et al., PloS One. 2013). The authors should revise the text accordingly.

Line 155: CCU needs more than 12h - How to store the cultures until transformation? How many plasmid was used in each transformation? The authors should add this information.

Line 182: What were the criteria to differentiate *M. hominis* infection from colonisation?

Line 230: "... which had the most highly conserved sequence..." What does this statement refer to?

Line 232ff: Why didn't the authors choose and analyse the RR of 6227 tetM?

Reviewer #2 (Comments for the Author):

The paper by Guiraud et al., describes a transformation protocol with 100-fold improved transformation efficiency compared with the previous best method, with capacity to transform the PG21 reference strain as well as clinical isolates, suggesting a wide application among *M. hominis* isolates. The use of a tet(M) gene from *M. hominis* instead of *E. faecalis* is a fascinating idea, and makes perfect sense to be used for the selection marker in conjunction with an improved promoter.

I only have a few minor comments relating to this manuscript. This new tool will be integral to understanding the virulence of *M. hominis*, a subject which is considerably lacking detail. A great deal of work has gone into constructing this library and the authors should be commended for their persistence in obtaining transformants.

Line 110: Suggest changing 'harbouring tet(M) were' to 'harbouring the tet(M) resistance gene'. This will make it clearer what tet(M) is for those that are not familiar with AMR among mycoplasmas.

Line 117: Suggest changing this sentence to something like the following 'Sequencing of PCR amplicons was performed' as you do not sequence the 'reaction'.

Line 150: What was the source of the 10^4 CFU/ml of *M. hominis*? Was this from a frozen -80 oC stock which was at a known concentration? If so, please indicate.

Line 151: Are you able to comment on the container which the cultures were grown in and the volume of culture used? Was this in a tissue culture flask in normoxic atmospheric conditions? Or in a screw top vessel preventing gas exchange with the surrounding environment?

Line 149 - 157: Mycoplasmas hominis transformation protocol section - You give conditions of how you grew the *M. hominis*, but I do not see any detail on the step in which the transformation occurs.

Line 154: You state 'Bacterial titers were evaluated before transformation by determining color changing units (CCUs)'. I am confused by this sentence. If you were growing the cells to mid-log phase and then evaluating the number of CCU before transformation, you would need to wait 48 hours until you had your CCU reading at which point your culture will have been well past log phase. Or is it that you are stating you set up the CCU assay before transforming the cells, and gained the result from this when the CCU a few days later and then used this to help with calculate the transformation efficiency?

Line 182: Suggest 'Serum from a patient with a *M. hominis* infection'

Line 182: What sort of infection did this patient have? Was it a true infection, or positive detection in a patient which was colonised by this organism? In figure 3, you show cleavage of both IgG and IgM suggesting this was a recent acquisition of *M. hominis* (IgM response), but also long-lasting antibody immunity (IgG). Did the patient have a previous history of *M. hominis* infection? Please also include a statement regarding ethics and the use of the serum from this patient.

Line 183: What was the reason for the albumin depletion?

Line 187: 1.108 do you mean 1×10^8

Line 330: This sentence seems to be partial 'efficient *M.*'

Line 341 & 342: Please give the full genus name for bacteria on the first use

Staff Comments:

Preparing Revision Guidelines

Please return the manuscript within 60 days; if you cannot complete the modification within this time period, please contact me. If you do not wish to modify the manuscript and prefer to submit it to another journal, please notify me of your decision immediately so that the manuscript may be formally withdrawn from consideration by Microbiology Spectrum.

Reviewer comments:

Reviewer #1 (Comments for the Author):

The manuscript of Guiraud et al. deals with the description of an improved transformation efficiency of *Mycoplasma hominis* using a genetically modified *tetM*-plasmid and the characterization of a respective disruption of the MIB-MIP system targeting human immunoglobulins.

There are these two main topics, genetics for an improved transformation efficiency and characterization of the MIB-MIP system disruption, whose combination led to the question, what the goal of the study was.

You're right that our manuscript is divided into two parts. The first one, which is the main goal of the study, is dedicated to the improvement of the transformation efficiency of *M. hominis* which has been very difficult to transform until this work. The second part is a proof of concept that transposon mutagenesis allows gene disruption and may enable the identification of putative virulence factors in *M. hominis*.

We specified these points in the introduction of the revised manuscript, lines 93-99.

The following comments may help to make the manuscript more complete on one side or the other.

Major comments1. For better understanding of the transformation part, a table should be added comprising ALL transformation experiments described in the results with columns of 1. transformed *M. hominis* strain/ 2. plasmid-name/ 3. RR-version / 4. *tetM*-version/origin / 5. number of successful to total transformations / 6. number of transformants per experiment / 7. transformation efficiency. Is it possible to calculate significance for transformation efficiency per plasmid ?

As suggested, we designed a table (Table 1) that summarized all transformation experiments (see below). In addition, comparisons of transformation efficiencies per plasmid for the *M. hominis* M132 strain were calculated using the Student test (one-tailed *t*-test, two samples unequal variance, heteroscedastic) and no significant difference of transformation efficiency was observed.

Table 1. Transformation experiments performed in this study.

Transformed M. hominis strain	Plasmid name	RR ^a -version	tet(M) origin	Number of successful to total transformations	Mean number of transformants per experiment	Transformation efficiency	Significance ^b (t -test)
M132	pMT85-Tet	Spiralin promoter	E. faecalis	3/6	1	4.8×10^{-9}	
M132	pMT85-PS-tetMMho	Spiralin promoter	M. hominis	0/4	0	-	
M132	pMT85-PAD-tetMMho	M. hominis arginine deiminase gene RR	M. hominis	0/3	0	-	
M132	pMT85-PtufA-tetMMho	M. hominis elongation factor gene RR	M. hominis	6/6	3	3.5×10^{-9}	$p = 0.39$
M132	pMT85-SynMyco-tetMMho	SynMyco RR	M. hominis	6/6	120	7.7×10^{-7}	$p = 0.10$
PG21	pMT85-SynMyco-tetMMho	SynMyco RR	M. hominis	2/2	36	4.7×10^{-7}	
4788	pMT85-SynMyco-tetMMho	SynMyco RR	M. hominis	1/1	2	1.3×10^{-8}	
5012	pMT85-SynMyco-tetMMho	SynMyco RR	M. hominis	1/1	4	4.2×10^{-9}	
4016	pMT85-SynMyco-tetMMho	SynMyco RR	M. hominis	1/1	6	2.2×10^{-9}	

^aRR ; regulatory region

^bOne-tailed *t*-test *p*-values are indicated for transformation efficiency obtained with the pMT85-PtufA-tetMMho and the pMT85-SynMyco-tetMMho plasmids compared to the transformation efficiency obtained with pMT85-Tet plasmid.

For better understanding of the transformation experiment part, we mentioned Table 1 in the results section, lines 261 and 274 of the revised manuscript.

2. Why do the authors focus on strain M132 (as reference) and not PG21 (as type strain)? It is very difficult to follow and comprehend the transposon insertion sites without having access to an ncbi-deposited genome sequence with annotated genes and proteins with postulated functions. The BioSample accession number SAMN33331381 and BioProject PRJNA493181 do neither contain the genome sequence nor genes or proteins, and using MolliGen I was unable to find the genes and encoded proteins described in a simple way. The same transformation analysis was done with PG21, a well-characterized type strain on genetic and protein level with the further advantage of carrying less mobile genetic elements and nearly representing a core-genome.

We agree with reviewer 1 that *M. hominis* strain PG21 is a well-characterized type strain at the genetic and protein level. However, the first transformation protocol was developed in our laboratory for the M132 strain only (Rideau *et al*, 2019). Despite several transformation attempts using the plasmid pMT85-Tet, the PG21 strain could never be transformed before the development of the SynMyco-based construction in the present study. This is why we focused on the M132 strain.

A comprehensive bioinformatic analysis and annotation have not been performed for the data provided in the BioSample accession number SAMN33331381. Nevertheless, for a better presentation of the transposon insertion sites, we added all PG21 homologs in Table S2. In addition, we made the genome sequence of the *M. hominis* strain M132 publicly available in the MolliGen database, so the data regarding the genes and encoded proteins described in the manuscript can now be easily found.

Thus, without having a comfortable way to get information about the affected M132 genes, it would be necessary to insert a column to the table of transposon insertion sites with accession numbers of the homologs and, in case of PG21 homologs, the MHO_XXXX numbers in addition.

As suggested, we inserted a column in the table of transposon insertion sites (Table S2) with accession numbers of the homologs. In case of PG21 homologs, the MHO_XXXX mnemonic was added in a separate column for a better readability of the table.

3. In Figure S1 the reverse complementary strand is given but also ending with ATG. Is it correct and represents the start codon of an upstream positioned gene on that strand? However, it is not correct to entitle the reverse complementary strand with the sequence orientation of 3'-5'. It is always 5'-3' and should be entitled as "reverse complementary strand".

Yes, the reverse complementary strand is ending with ATG, which is the start codon of an upstream gene positioned on that strand (MHO_0680). We have already mentioned in the result section of the original manuscript that "the arginine deiminase gene and the adjacent gene encoding a hypothetical protein were located on the opposite-sense strand". However, we made an error in Figure S1 as it was not 3'-5' but, as you noticed, 5'-3'. As suggested, we also added the "reverse complementary strand" title.

4. Multiple sequence alignments of RR- and tetM-variants should replace the display of individual sequences of the same region/ gene part in different species. Differences in *M. hominis* and

Enterococcus tetM should be shown (line 220 and Fig.S2/3). If homology of RR regions is too low, a multiple scheme alignment of RR-composition (sequence of (TGC)-Xn-(TAWAAT)-Xn-.... - (TGC)-Xn-(TAWAAT)-Xn - (ATG)) per species/variant would be informative and even here the acc-number of the sequences should be given with the information about the part of the region shown.

Multiple sequence alignments of *tetM* variants is displayed in Table S2, in which differences between several *M. hominis* isolates and *Enterococcus tetM* are shown. As suggested and to clarify this point, we specified the origin of the *tet(M)* gene carried by the original plasmid in the result section, line 233 of the revised manuscript.

As suggested, we added in Figure S1, a multiple scheme alignment of the RR composition of the regions upstream of the Tu and arginine deiminase genes, and of the SynMyco RR (Figure S1D). In addition, as suggested, the accession number of the sequences and the localization of the regions displayed were added in Figure S1.

Is it known, whether sequences further upstream than the ATG-proximal (TGC)-Xn (TAWAAT) motifs are involved in gene regulation? Why were the full intergenic sequences displayed?

To our knowledge, sequences further upstream than the ATG-proximal (TGC)-Xn (TAWAAT) motifs may be involved in gene regulation. For example, in a previous study (Krásný L. *et al.* J Bacteriol. 2000), the authors reported that the strength of the *tuf* promoter to initiate transcription is about 20-fold higher than that of the *str* operon promoter in *Bacillus* sp. Then, the authors explained that the different strengths of the promoters were the consequence of a combined effect of oppositely acting *cis* elements, identified upstream of *str* promoter (an inhibitory region) and *tuf* promoter (a stimulatory A/T-rich block). Consequently, we cannot exclude that regions upstream of the gene promoter may have some influence on the gene expression. This is why we selected the entire intergenic region upstream of the Tu elongation factor gene. However, according to the results of Montero-Blay *et al.*, we previously suggested in the discussion of the original manuscript that the “use of a shorter sequence for the RRs of the arginine deiminase and *tuf* genes could further increase the transformation efficiency”.

We explained our reasoning and quoted the reference (Krásný L. *et al.* J Bacteriol. 2000) in the discussion, lines 346-348 of the revised manuscript. In addition, we referred to the multiple scheme alignment of the shorter regulatory region sequences displayed in figure S1D, in the discussion section, line 362 of the revised manuscript.

5. Figures S2 and S3 could be fused with nucleotide sequences given as consensus with wobble IUPAC-IUB symbols and the consensus amino acid sequence above with an indication of isofunctional amino acid changes. That would be sufficient to see variabilities in gene and protein sequences.

As suggested, Figures S2 and S3 were fused. The consensus sequences were displayed above with an indication of nucleotide and amino acid changes.

In addition, for a better interpretation of the figure S2, we added two sentences in the legends of the updated file “The consensus nucleotide sequence corresponding to that of the *Enterococcus*-derived pMT85-Tet plasmid is indicated above and indication of single nucleotide substitutions is provided below (n=30)” and “The consensus amino acid sequence corresponding to that of the *Enterococcus*-derived pMT85-Tet plasmid is indicated above and indication of amino acid changes is provided below (n=13).”

Non-isofunctional amino acid changes should be discussed in the text, as their impact on the protein structure will be higher than that of the isofunctional ones.

As suggested, we performed prediction of the possible impact of the non-isofunctional amino acid changes using PolyPhen-2. Although 5 amino acid substitutions linked to charge or hydrophobicity differed between the *M. hominis* Tet(M) sequence and that of the pMT85-Tet (namely H214Q, P251Q, N257K, K265E and S285L), the analysis revealed a probable minor impact of these amino acid changes on the protein structure.

The sentence “Indeed, the prediction of the possible impact of these five amino acid substitutions (namely H214Q, P251Q, N257K, K265E and S285L) using PolyPhen-2 revealed a probable minor impact on the protein structure, explaining the absence of clear difference in transformation efficiency observed between both conditions (10^{-9} vs $<10^{-9}$).” was added in the discussion, lines 333-337 of the revised manuscript.

6. What was the rationale to characterise a more rare insertion event? Was it done as "proof-of-principle"? Only three times the MIB-MIP system was affected, of which only one was further characterized. The demonstration of MIB-MIP-dependent loss of human immunoglobulin cleavage is good, but is this sufficient to demonstrate the utility of an improved, but still random, transposon insertion as a tool for detecting and characterizing virulence genes?

You're right that we elected to characterize a transformant which integrated the transposon in a region with a relatively low insertion event frequency. However, as explained in comment #1, the characterization of this transformant is a proof of principle that transposon mutagenesis allows gene disruption and may thus enable the characterization of putative virulence factors in *M. hominis*. In addition, we already had a phenotypic assay, easy to set up, with a clear demonstration of functionality (ON/OFF behavior) for this MIB-MIP system. However, we agree that the development of a targeted mutagenesis method would even be more efficient to detect and characterize virulence genes. This will be the aim of further studies.

This point was specified in the discussion, lines 417-419 of the revised manuscript.

However, it looks like as if transposon integration dominantly takes place in hotspots of variant lipoproteins, MGEs, defence islands etc. and will thus have the potential to rather destroy virulence-associated structures than core genes. Would it strengthen the line of the manuscript in mapping the transposon insertion sites to the *M. hominis* (PG21) genome and to analyse whether core gene regions are less affected than more variant (hotspot-like) regions? Would it be possible to calculate significances between both groups?

You're right that it looks like that transposon integration may dominantly take place in hotspots of variant lipoproteins and MGE. However, further transformation experiments as well as the development of a more efficient method to identify the insertion sites are necessary to determine whether core gene regions may be less affected than more variable (hotspot-like) regions. Here, the number of characterized insertion sites still remain too limited to conclude. As specified in comment #2, in case of insertion event in PG21 homologs, the MHO_XXXX mnemonic was added in Table S2. In addition, one can assume that “core genes” have critical functions and are likely to be essential. Using such a transposon mutagenesis method, no insertion events in essential loci can be recovered. Thus, it would be not suitable to calculate here significances between insertion events in core gene regions and in accessory gene regions. We specified these points in the discussion of the revised manuscript, lines 402-404.

7. Minor comments

Line 61f: What does the authors want to express with "...colonization to invasiveness is not

established...?"

Invasiveness of *M. hominis* in cells is rarely, but well documented (e.g. Taylor-Robinson et al., International journal of experimental pathology. 1991; Gdoura et al., BMC infectious diseases. 2007; Diaz-Garcia et al., Hum Reprod. 2006; Hopfe et al., PloS One. 2013). The authors should revise the text accordingly.

Sorry our sentence was unclear. We meant "infection" and not "invasiveness", which has a different meaning. The sentence lines 61-62 of the revised manuscript was modified accordingly.

Line 155: CCU needs more than 12h - How to store the cultures until transformation? How many plasmid was used in each transformation? The authors should add this information.

We set up the CCU numbering assay before transforming the cells and gained the result from this assay a few days later. The result was then used to calculate the transformation efficiencies. Thus, the transformation was performed without being aware of the result of the CCU assay. This point was specified in the material and method section, lines 164-167 of the revised manuscript with the sentence : "Bacterial titer was evaluated before transforming the cells by determining color changing units (CCUs) and CFUs/mL as described previously (4). The result of this assay was gained a few days after the transformation experiment and was used to calculate the transformation efficiency."

The *M. hominis* strains were transformed with 10 µg of methylated plasmid. This quantity of plasmid was added, lines 159-160 of the revised manuscript.

Line 182: What were the criteria to differentiate *M. hominis* infection from colonisation?

In our case, *M. hominis* was responsible for a post-partum endometritis. It was detected by culture in the patient's placenta and also in the peripheral samples of her newborn. The clinical context and the isolation of *M. hominis* in these specimens with a high load (10^4 CCU/ml), are criteria to confirm a true *M. hominis* infection. To specify this point, data regarding the clinical context were added lines 193-194 in the revised manuscript.

Line 230: "... which had the most highly conserved sequence..." What does this statement refer to?

Sorry, the sentence is unclear. Given that the Tet(M) protein sequence of the *M. hominis* 6227 harbored amino acid changes shared by most *M. hominis* analysed isolates, we considered this sequence as a consensus and selected it for further development. This point was clarified lines 240-242 of the revised manuscript.

Line 232ff: Why didn't the authors choose and analyse the RR of 6227 tetM?

We agree with the reviewer 1 that the choice of the RR of 6227 *tet(M)* to drive the tetracycline resistance gene would be a good idea. This idea has already been suggested in the discussion of the original manuscript with the sentence: "Alternatively, the native RR of the antibiotic resistance marker could be added to the plasmid". Indeed, in this study, we first decided to focus on the regulatory regions of genes with high transcription level in the hope of improving the expression of the *tet(M)* resistance gene.

The discussion section was specified by adding the following sentence: "For example, the impact of the 6227 *tet(M)* gene RR on the transformation efficiency could be evaluated in further attempts." lines 366-367 of the revised manuscript.

Reviewer #2 (Comments for the Author):

The paper by Guiraud *et al.*, describes a transformation protocol with 100-fold improved transformation efficiency compared with the previous best method, with capacity to transform the PG21 reference strain as well as clinical isolates, suggesting a wide application among *M. hominis* isolates. The use of a *tet(M)* gene from *M. hominis* instead of *E. faecalis* is a fascinating idea, and makes perfect sense to be used for the selection marker in conjunction with an improved promoter.

I only have a few minor comments relating to this manuscript. This new tool will be integral to understanding the virulence of *M. hominis*, a subject which is considerably lacking detail. A great deal of work has gone into constructing this library and the authors should be commended for their persistence in obtaining transformants.

Thank you for these very nice comments.

Line 110: Suggest changing 'harbouring tet(M) were' to 'harbouring the tet(M) resistance gene'. This will make it clearer what tet(M) is for those that are not familiar with AMR among mycoplasmas.

As suggested, we modified the sentence line 111 of the revised manuscript.

Line 117: Suggest changing this sentence to something like the following 'Sequencing of PCR amplicons was performed' as you do not sequence the 'reaction'.

Modified as suggested, line 119 of the revised manuscript.

Line 150: What was the source of the 10^4 CFU/ml of *M. hominis*? Was this from a frozen -80°C stock which was at a known concentration? If so, please indicate.

Yes, a frozen -80°C stock culture of *M. hominis* at a known concentration was thawed and diluted to obtain 10^4 CFU/mL. We specified this point in the materials and methods section, lines 151-153 of the revised manuscript.

Line 151: Are you able to comment on the container which the cultures were grown in and the volume of culture used? Was this in a tissue culture flask in normoxic atmospheric conditions? Or in a screw top vessel preventing gas exchange with the surrounding environment?

The cultures were grown in sealed hemolysis tubes containing 10 mL of HA medium. The cultures were incubated at 37°C without CO_2 until they reached the mid-logarithmic phase.

As suggested, we added these points in the revised manuscript, lines 153-154.

Line 149 - 157: *Mycoplasmas hominis* transformation protocol section - You give conditions of how you grew the *M. hominis*, but I do not see any detail on the step in which the transformation occurs.

You're right, given that all details on the transformation protocol have already been provided in Rideau *et al.*, ACS Synth Biol, 2019, only the modified parameters have been initially mentioned, namely the pre-culture conditions.

After inoculation of *M. hominis* in HA medium and obtention of the mid-log phase, “Cells were harvested by centrifugation at 10,000 g for 20 min at 4°C and the pellet was washed twice with Tris Buffer. Cells were then suspended in CaCl₂ and incubated at 4 °C for 30 min. CaCl₂-incubated cells (100 µL) were gently mixed with 10 µg of yeast tRNA (Life Technologies) and 10 µg of methylated plasmid. This mixture was poured onto 1.5 mL of 40% PEG 8000 (Sigma-Aldrich) for 30 min at room temperature then 7.5 mL of HA liquid medium were added before incubation for 3 hours at 37 °C. After centrifugation at 8,000 g for 10 min, the pellet was suspended in 1 mL of HA liquid medium and plated onto HA agar supplemented with 2 µg/mL tetracycline and incubated at 37°C in 5% CO₂”.

As suggested, for better understanding when the transformation occurs, the previous paragraph was added in the material and method section, lines 156-164 of the revised manuscript.

Line 154: You state 'Bacterial titers were evaluated before transformation by determining color changing units (CCUs)'. I am confused by this sentence. If you were growing the cells to mid-log phase and then evaluating the number of CCU before transformation, you would need to wait 48 hours until you had your CCU reading at which point your culture will have been well past log phase. Or is it that you are stating you set up the CCU assay before transforming the cells, and gained the result from this when the CCU a few days later and then used this to help with calculate the transformation efficiency?

You're right, we set up the CCU assay before transforming the cells, and gained the result from this a few days later. We then used this result to calculate the transformation efficiency. Thus, the transformation was performed without being aware of the result of the CCU assay.

To clarify this point, we added the sentence “The result of this assay was gained a few days after the transformation experiment and was used to calculate the transformation efficiency.” lines 166-167 of the revised manuscript.

Line 182: Suggest 'Serum from a patient with a *M. hominis* infection'

The sentence was modified as suggested line 192 of the revised manuscript.

Line 182: What sort of infection did this patient have? Was it a true infection, or positive detection in a patient which was colonised by this organism? In figure 3, you show cleavage of both IgG and IgM suggesting this was a recent acquisition of *M. hominis* (IgM response), but also long-lasting antibody immunity (IgG). Did the patient have a previous history of *M. hominis* infection? Please also include a statement regarding ethics and the use of the serum from this patient.

The patient had a true *M. hominis* infection. Indeed, *M. hominis* was responsible for a post-partum endometritis. *M. hominis* was isolated by culture from the patient's placenta and also from the peripheral samples of her newborn. Remnant serum from the mother was anonymously collected two days after childbirth in the context of routine clinical care, explaining the presence of an IgM response in the Western blot. There was no information about a previous history of *M. hominis* infection.

As suggested, we specified the clinical context of the serum collection, lines 192-194 of the revised manuscript.

Informed consent is not required in France for the use of remnants of microbiological samples in the quality assurance of diagnostic methods according to the national legislation. The collection and use of anonymized samples were approved by the French Personal Data Protection Authority (CNIL, n°10.362). As suggested, the statement regarding ethics for the use of the patient serum was provided in the “Informed Consent Statement” section, lines 648-652 of the revised manuscript.

Line 183: What was the reason for the albumin depletion?

Serum was depleted of albumin because albumin has a molecular weight of 68 kDa, which is close to that of the IgG heavy chain (55 kDa intact, 44 cut) and IgM heavy chain (70 kDa, 59 cut). Albumin is highly abundant in serum, and consequently, is the largest protein band visible when performing a SDS-PAGE on a serum sample. As a result, the large albumin band has a tendency to deform and mask the much less abundant IgG/IgM bands (Nottelet *et al.*, Sci Adv, 2021). We therefore preferred to lower the albumin concentration through depletion in order to have a cleaner IgG/IgM band.

As suggested, we added these clarifications and the reference 27 (Nottelet *et al.*, Sci Adv, 2021) in the material and method section, lines 194-197 of the revised manuscript.

Line 187: 1.108 do you mean 1×10^8

You're right, we modified this point in the revised manuscript, line 200, as suggested.

Line 330: This sentence seems to be partial 'efficient M.'

Sorry, the sentence was incomplete. The complete sentence "However, the chosen 212 bp putative regulatory region including the main sequence determinants (Figure S1) was not suitable for efficient *M. hominis* transformation" was specified in the revised manuscript, line 342-343.

Line 341 & 342: Please give the full genus name for bacteria on the first use

As suggested, we mentioned the full genus name of bacteria in the discussion, lines 357-358 of the revised manuscript.

July 11, 2023

Prof. Sabine Pereyre
Univ. Bordeaux, CNRS
UMR 5234 Fundamental Microbiology and Pathogenicity
146 rue Léo Saignat
Bordeaux 33076
France

Re: Spectrum01873-23R1 (Improved transformation efficiency in *Mycoplasma hominis* enables disruption of the MIB-MIP system targeting human immunoglobulins)

Dear Prof. Sabine Pereyre:

Thank you for submitting your manuscript to Microbiology Spectrum. As you will see your paper is very close to acceptance. Please modify the manuscript along the lines recommended by reviewer #1. As these revisions are quite minor, I expect that you should be able to turn in the revised paper in less than 30 days, if not sooner. If your manuscript was reviewed, you will find the reviewers' comments below.

When submitting the revised version of your paper, please provide (1) point-by-point responses to the issues raised by the reviewers as file type "Response to Reviewers," not in your cover letter, and (2) a PDF file that indicates the changes from the original submission (by highlighting or underlining the changes) as file type "Marked Up Manuscript - For Review Only". Please use this link to submit your revised manuscript. Detailed instructions on submitting your revised paper are below.

Link Not Available

Sincerely,

Olivier Neyrolles, PhD

Reviewer comments:

Reviewer #1 (Comments for the Author):

The authors have answered the comments satisfactorily, so that in the manuscript the few passages that were difficult to understand have disappeared.

They have improved the traceability of their data.

However, in supplementary table 2, column K, the accession number of the homologs refers to the genome sequence of the isolate, the homolog protein given in column J derived from. It would be easy to specify the accession number of each protein (e.g. in replacing "MHO_0350 in NC_013511.1" by WP_012855330.1) In my mind, easy accessibility of data will facilitate scientific exchange and discussion.

If still possible the authors should use a non-proportional font in the multiple sequence alignment of Fig. S1 D.

Reviewer #2 (Comments for the Author):

Thank you for addressing the points raised. This is very interesting work and I look forward to seeing it published.

Preparing Revision Guidelines

Please return the manuscript within 60 days; if you cannot complete the modification within this time period, please contact me. If you do not wish to modify the manuscript and prefer to submit it to another journal, please notify me of your decision immediately so that the manuscript may be formally withdrawn from consideration by Microbiology Spectrum.

Reviewer comments:

Reviewer #1 (Comments for the Author):

The manuscript of Guiraud et al. deals with the description of an improved transformation efficiency of *Mycoplasma hominis* using a genetically modified *tetM*-plasmid and the characterization of a respective disruption of the MIB-MIP system targeting human immunoglobulins.

There are these two main topics, genetics for an improved transformation efficiency and characterization of the MIB-MIP system disruption, whose combination led to the question, what the goal of the study was.

You're right that our manuscript is divided into two parts. The first one, which is the main goal of the study, is dedicated to the improvement of the transformation efficiency of *M. hominis* which has been very difficult to transform until this work. The second part is a proof of concept that transposon mutagenesis allows gene disruption and may enable the identification of putative virulence factors in *M. hominis*.

We specified these points in the introduction of the revised manuscript, lines 93-99.

The following comments may help to make the manuscript more complete on one side or the other.

Major comments1. For better understanding of the transformation part, a table should be added comprising ALL transformation experiments described in the results with columns of 1. transformed *M. hominis* strain/ 2. plasmid-name/ 3. RR-version / 4. *tetM*-version/origin / 5. number of successful to total transformations / 6. number of transformants per experiment / 7. transformation efficiency. Is it possible to calculate significance for transformation efficiency per plasmid ?

As suggested, we designed a table (Table 1) that summarized all transformation experiments (see below). In addition, comparisons of transformation efficiencies per plasmid for the *M. hominis* M132 strain were calculated using the Student test (one-tailed *t*-test, two samples unequal variance, heteroscedastic) and no significant difference of transformation efficiency was observed.

Table 1. Transformation experiments performed in this study.

Transformed M. hominis strain	Plasmid name	RR ^a -version	tet(M) origin	Number of successful to total transformations	Mean number of transformants per experiment	Transformation efficiency	Significance ^b (t -test)
M132	pMT85-Tet	Spiralin promoter	E. faecalis	3/6	1	4.8×10^{-9}	
M132	pMT85-PS-tetMMho	Spiralin promoter	M. hominis	0/4	0	-	
M132	pMT85-PAD-tetMMho	M. hominis arginine deiminase gene RR	M. hominis	0/3	0	-	
M132	pMT85-PtufA-tetMMho	M. hominis elongation factor gene RR	M. hominis	6/6	3	3.5×10^{-9}	$p = 0.39$
M132	pMT85-SynMyco-tetMMho	SynMyco RR	M. hominis	6/6	120	7.7×10^{-7}	$p = 0.10$
PG21	pMT85-SynMyco-tetMMho	SynMyco RR	M. hominis	2/2	36	4.7×10^{-7}	
4788	pMT85-SynMyco-tetMMho	SynMyco RR	M. hominis	1/1	2	1.3×10^{-8}	
5012	pMT85-SynMyco-tetMMho	SynMyco RR	M. hominis	1/1	4	4.2×10^{-9}	
4016	pMT85-SynMyco-tetMMho	SynMyco RR	M. hominis	1/1	6	2.2×10^{-9}	

^aRR ; regulatory region

^bOne-tailed *t*-test *p*-values are indicated for transformation efficiency obtained with the pMT85-PtufA-tetMMho and the pMT85-SynMyco-tetMMho plasmids compared to the transformation efficiency obtained with pMT85-Tet plasmid.

For better understanding of the transformation experiment part, we mentioned Table 1 in the results section, lines 261 and 274 of the revised manuscript.

2. Why do the authors focus on strain M132 (as reference) and not PG21 (as type strain)? It is very difficult to follow and comprehend the transposon insertion sites without having access to an ncbi-deposited genome sequence with annotated genes and proteins with postulated functions. The BioSample accession number SAMN33331381 and BioProject PRJNA493181 do neither contain the genome sequence nor genes or proteins, and using MolliGen I was unable to find the genes and encoded proteins described in a simple way. The same transformation analysis was done with PG21, a well-characterized type strain on genetic and protein level with the further advantage of carrying less mobile genetic elements and nearly representing a core-genome.

We agree with reviewer 1 that *M. hominis* strain PG21 is a well-characterized type strain at the genetic and protein level. However, the first transformation protocol was developed in our laboratory for the M132 strain only (Rideau *et al*, 2019). Despite several transformation attempts using the plasmid pMT85-Tet, the PG21 strain could never be transformed before the development of the SynMyco-based construction in the present study. This is why we focused on the M132 strain.

A comprehensive bioinformatic analysis and annotation have not been performed for the data provided in the BioSample accession number SAMN33331381. Nevertheless, for a better presentation of the transposon insertion sites, we added all PG21 homologs in Table S2. In addition, we made the genome sequence of the *M. hominis* strain M132 publicly available in the MolliGen database, so the data regarding the genes and encoded proteins described in the manuscript can now be easily found.

Thus, without having a comfortable way to get information about the affected M132 genes, it would be necessary to insert a column to the table of transposon insertion sites with accession numbers of the homologs and, in case of PG21 homologs, the MHO_XXXX numbers in addition.

As suggested, we inserted a column in the table of transposon insertion sites (Table S2) with accession numbers of the homologs. In case of PG21 homologs, the MHO_XXXX mnemonic was added in a separate column for a better readability of the table.

3. In Figure S1 the reverse complementary strand is given but also ending with ATG. Is it correct and represents the start codon of an upstream positioned gene on that strand? However, it is not correct to entitle the reverse complementary strand with the sequence orientation of 3'-5'. It is always 5'-3' and should be entitled as "reverse complementary strand".

Yes, the reverse complementary strand is ending with ATG, which is the start codon of an upstream gene positioned on that strand (MHO_0680). We have already mentioned in the result section of the original manuscript that "the arginine deiminase gene and the adjacent gene encoding a hypothetical protein were located on the opposite-sense strand". However, we made an error in Figure S1 as it was not 3'-5' but, as you noticed, 5'-3'. As suggested, we also added the "reverse-complementary strand" title.

4. Multiple sequence alignments of RR- and tetM-variants should replace the display of individual sequences of the same region/ gene part in different species. Differences in *M. hominis* and

Enterococcus tetM should be shown (line 220 and Fig.S2/3). If homology of RR regions is too low, a multiple scheme alignment of RR-composition (sequence of (TGC)-Xn-(TAWAAT)-Xn-.... - (TGC)-Xn-(TAWAAT)-Xn - (ATG)) per species/variant would be informative and even here the acc-number of the sequences should be given with the information about the part of the region shown.

Multiple sequence alignments of *tetM* variants is displayed in Table S2, in which differences between several *M. hominis* isolates and *Enterococcus tetM* are shown. As suggested and to clarify this point, we specified the origin of the *tet(M)* gene carried by the original plasmid in the result section, line 233 of the revised manuscript.

As suggested, we added in Figure S1, a multiple scheme alignment of the RR composition of the regions upstream of the Tu and arginine deiminase genes, and of the SynMyco RR (Figure S1D). In addition, as suggested, the accession number of the sequences and the localization of the regions displayed were added in Figure S1.

Is it known, whether sequences further upstream than the ATG-proximal (TGC)-Xn (TAWAAT) motifs are involved in gene regulation? Why were the full intergenic sequences displayed?

To our knowledge, sequences further upstream than the ATG-proximal (TGC)-Xn (TAWAAT) motifs may be involved in gene regulation. For example, in a previous study (Krásný L. *et al.* J Bacteriol. 2000), the authors reported that the strength of the *tuf* promoter to initiate transcription is about 20-fold higher than that of the *str* operon promoter in *Bacillus* sp. Then, the authors explained that the different strengths of the promoters were the consequence of a combined effect of oppositely acting *cis* elements, identified upstream of *str* promoter (an inhibitory region) and *tuf* promoter (a stimulatory A/T-rich block). Consequently, we cannot exclude that regions upstream of the gene promoter may have some influence on the gene expression. This is why we selected the entire intergenic region upstream of the Tu elongation factor gene. However, according to the results of Montero-Blay *et al.*, we previously suggested in the discussion of the original manuscript that the “use of a shorter sequence for the RRs of the arginine deiminase and *tuf* genes could further increase the transformation efficiency”.

We explained our reasoning and quoted the reference (Krásný L. *et al.* J Bacteriol. 2000) in the discussion, lines 346-348 of the revised manuscript. In addition, we referred to the multiple scheme alignment of the shorter regulatory region sequences displayed in figure S1D, in the discussion section, line 362 of the revised manuscript.

5. Figures S2 and S3 could be fused with nucleotide sequences given as consensus with wobble IUPAC-IUB symbols and the consensus amino acid sequence above with an indication of isofunctional amino acid changes. That would be sufficient to see variabilities in gene and protein sequences.

As suggested, Figures S2 and S3 were fused. The consensus sequences were displayed above with an indication of nucleotide and amino acid changes.

In addition, for a better interpretation of the figure S2, we added two sentences in the legends of the updated file “The consensus nucleotide sequence corresponding to that of the *Enterococcus*-derived pMT85-Tet plasmid is indicated above and indication of single nucleotide substitutions is provided below (n=30)” and “The consensus amino acid sequence corresponding to that of the *Enterococcus*-derived pMT85-Tet plasmid is indicated above and indication of amino acid changes is provided below (n=13).”

Non-isofunctional amino acid changes should be discussed in the text, as their impact on the protein structure will be higher than that of the isofunctional ones.

As suggested, we performed prediction of the possible impact of the non-isofunctional amino acid changes using PolyPhen-2. Although 5 amino acid substitutions linked to charge or hydrophobicity differed between the *M. hominis* Tet(M) sequence and that of the pMT85-Tet (namely H214Q, P251Q, N257K, K265E and S285L), the analysis revealed a probable minor impact of these amino acid changes on the protein structure.

The sentence “Indeed, the prediction of the possible impact of these five amino acid substitutions (namely H214Q, P251Q, N257K, K265E and S285L) using PolyPhen-2 revealed a probable minor impact on the protein structure, explaining the absence of clear difference in transformation efficiency observed between both conditions (10^{-9} vs $<10^{-9}$).” was added in the discussion, lines 333-337 of the revised manuscript.

6. What was the rationale to characterise a more rare insertion event? Was it done as "proof-of-principle"? Only three times the MIB-MIP system was affected, of which only one was further characterized. The demonstration of MIB-MIP-dependent loss of human immunoglobulin cleavage is good, but is this sufficient to demonstrate the utility of an improved, but still random, transposon insertion as a tool for detecting and characterizing virulence genes?

You're right that we elected to characterize a transformant which integrated the transposon in a region with a relatively low insertion event frequency. However, as explained in comment #1, the characterization of this transformant is a proof of principle that transposon mutagenesis allows gene disruption and may thus enable the characterization of putative virulence factors in *M. hominis*. In addition, we already had a phenotypic assay, easy to set up, with a clear demonstration of functionality (ON/OFF behavior) for this MIB-MIP system. However, we agree that the development of a targeted mutagenesis method would even be more efficient to detect and characterize virulence genes. This will be the aim of further studies.

This point was specified in the discussion, lines 417-419 of the revised manuscript.

However, it looks like as if transposon integration dominantly takes place in hotspots of variant lipoproteins, MGEs, defence islands etc. and will thus have the potential to rather destroy virulence-associated structures than core genes. Would it strengthen the line of the manuscript in mapping the transposon insertion sites to the *M. hominis* (PG21) genome and to analyse whether core gene regions are less affected than more variant (hotspot-like) regions? Would it be possible to calculate significances between both groups?

You're right that it looks like that transposon integration may dominantly take place in hotspots of variant lipoproteins and MGE. However, further transformation experiments as well as the development of a more efficient method to identify the insertion sites are necessary to determine whether core gene regions may be less affected than more variable (hotspot-like) regions. Here, the number of characterized insertion sites still remain too limited to conclude. As specified in comment #2, in case of insertion event in PG21 homologs, the MHO_XXXX mnemonic was added in Table S2. In addition, one can assume that “core genes” have critical functions and are likely to be essential. Using such a transposon mutagenesis method, no insertion events in essential loci can be recovered. Thus, it would be not suitable to calculate here significances between insertion events in core gene regions and in accessory gene regions. We specified these points in the discussion of the revised manuscript, lines 402-404.

7. Minor comments

Line 61f: What does the authors want to express with "...colonization to invasiveness is not

established..”?

Invasiveness of *M. hominis* in cells is rarely, but well documented (e.g. Taylor-Robinson et al., International journal of experimental pathology. 1991; Gdoura et al., BMC infectious diseases. 2007; Diaz-Garcia et al., Hum Reprod. 2006; Hopfe et al., PloS One. 2013). The authors should revise the text accordingly.

Sorry our sentence was unclear. We meant “infection” and not “invasiveness”, which has a different meaning. The sentence lines 61-62 of the revised manuscript was modified accordingly.

Line 155: CCU needs more than 12h - How to store the cultures until transformation? How many plasmid was used in each transformation? The authors should add this information.

We set up the CCU numbering assay before transforming the cells and gained the result from this assay a few days later. The result was then used to calculate the transformation efficiencies. Thus, the transformation was performed without being aware of the result of the CCU assay. This point was specified in the material and method section, lines 164-167 of the revised manuscript with the sentence : “Bacterial titer was evaluated before transforming the cells by determining color changing units (CCUs) and CFUs/mL as described previously (4). The result of this assay was gained a few days after the transformation experiment and was used to calculate the transformation efficiency.”

The *M. hominis* strains were transformed with 10 µg of methylated plasmid. This quantity of plasmid was added, lines 159-160 of the revised manuscript.

Line 182: What were the criteria to differentiate *M. hominis* infection from colonisation?

In our case, *M. hominis* was responsible for a post-partum endometritis. It was detected by culture in the patient’s placenta and also in the peripheral samples of her newborn. The clinical context and the isolation of *M. hominis* in these specimens with a high load (10^4 CCU/ml), are criteria to confirm a true *M. hominis* infection. To specify this point, data regarding the clinical context were added lines 193-194 in the revised manuscript.

Line 230: "... which had the most highly conserved sequence..." What does this statement refer to?

Sorry, the sentence is unclear. Given that the Tet(M) protein sequence of the *M. hominis* 6227 harbored amino acid changes shared by most *M. hominis* analysed isolates, we considered this sequence as a consensus and selected it for further development. This point was clarified lines 240-242 of the revised manuscript.

Line 232ff: Why didn't the authors choose and analyse the RR of 6227 tetM?

We agree with the reviewer 1 that the choice of the RR of 6227 *tet(M)* to drive the tetracycline resistance gene would be a good idea. This idea has already been suggested in the discussion of the original manuscript with the sentence: “Alternatively, the native RR of the antibiotic resistance marker could be added to the plasmid”. Indeed, in this study, we first decided to focus on the regulatory regions of genes with high transcription level in the hope of improving the expression of the *tet(M)* resistance gene.

The discussion section was specified by adding the following sentence: “For example, the impact of the 6227 *tet(M)* gene RR on the transformation efficiency could be evaluated in further attempts.” lines 366-367 of the revised manuscript.

Reviewer #2 (Comments for the Author):

The paper by Guiraud *et al.*, describes a transformation protocol with 100-fold improved transformation efficiency compared with the previous best method, with capacity to transform the PG21 reference strain as well as clinical isolates, suggesting a wide application among *M. hominis* isolates. The use of a *tet(M)* gene from *M. hominis* instead of *E. faecalis* is a fascinating idea, and makes perfect sense to be used for the selection marker in conjunction with an improved promoter.

I only have a few minor comments relating to this manuscript. This new tool will be integral to understanding the virulence of *M. hominis*, a subject which is considerably lacking detail. A great deal of work has gone into constructing this library and the authors should be commended for their persistence in obtaining transformants.

Thank you for these very nice comments.

Line 110: Suggest changing 'harbouring tet(M) were' to 'harbouring the tet(M) resistance gene'. This will make it clearer what tet(M) is for those that are not familiar with AMR among mycoplasmas.

As suggested, we modified the sentence line 111 of the revised manuscript.

Line 117: Suggest changing this sentence to something like the following 'Sequencing of PCR amplicons was performed' as you do not sequence the 'reaction'.

Modified as suggested, line 119 of the revised manuscript.

Line 150: What was the source of the 10^4 CFU/ml of *M. hominis*? Was this from a frozen -80°C stock which was at a known concentration? If so, please indicate.

Yes, a frozen -80°C stock culture of *M. hominis* at a known concentration was thawed and diluted to obtain 10^4 CFU/mL. We specified this point in the materials and methods section, lines 151-153 of the revised manuscript.

Line 151: Are you able to comment on the container which the cultures were grown in and the volume of culture used? Was this in a tissue culture flask in normoxic atmospheric conditions? Or in a screw top vessel preventing gas exchange with the surrounding environment?

The cultures were grown in sealed hemolysis tubes containing 10 mL of HA medium. The cultures were incubated at 37°C without CO_2 until they reached the mid-logarithmic phase.

As suggested, we added these points in the revised manuscript, lines 153-154.

Line 149 - 157: *Mycoplasmas hominis* transformation protocol section - You give conditions of how you grew the *M. hominis*, but I do not see any detail on the step in which the transformation occurs.

You're right, given that all details on the transformation protocol have already been provided in Rideau *et al.*, ACS Synth Biol, 2019, only the modified parameters have been initially mentioned, namely the pre-culture conditions.

After inoculation of *M. hominis* in HA medium and obtention of the mid-log phase, “Cells were harvested by centrifugation at 10,000 g for 20 min at 4°C and the pellet was washed twice with Tris Buffer. Cells were then suspended in CaCl₂ and incubated at 4 °C for 30 min. CaCl₂-incubated cells (100 µL) were gently mixed with 10 µg of yeast tRNA (Life Technologies) and 10 µg of methylated plasmid. This mixture was poured onto 1.5 mL of 40% PEG 8000 (Sigma-Aldrich) for 30 min at room temperature then 7.5 mL of HA liquid medium were added before incubation for 3 hours at 37 °C. After centrifugation at 8,000 g for 10 min, the pellet was suspended in 1 mL of HA liquid medium and plated onto HA agar supplemented with 2 µg/mL tetracycline and incubated at 37°C in 5% CO₂”.

As suggested, for better understanding when the transformation occurs, the previous paragraph was added in the material and method section, lines 156-164 of the revised manuscript.

Line 154: You state 'Bacterial titers were evaluated before transformation by determining color changing units (CCUs)'. I am confused by this sentence. If you were growing the cells to mid-log phase and then evaluating the number of CCU before transformation, you would need to wait 48 hours until you had your CCU reading at which point your culture will have been well past log phase. Or is it that you are stating you set up the CCU assay before transforming the cells, and gained the result from this when the CCU a few days later and then used this to help with calculate the transformation efficiency?

You're right, we set up the CCU assay before transforming the cells, and gained the result from this a few days later. We then used this result to calculate the transformation efficiency. Thus, the transformation was performed without being aware of the result of the CCU assay.

To clarify this point, we added the sentence “The result of this assay was gained a few days after the transformation experiment and was used to calculate the transformation efficiency.” lines 166-167 of the revised manuscript.

Line 182: Suggest 'Serum from a patient with a *M. hominis* infection'

The sentence was modified as suggested line 192 of the revised manuscript.

Line 182: What sort of infection did this patient have? Was it a true infection, or positive detection in a patient which was colonised by this organism? In figure 3, you show cleavage of both IgG and IgM suggesting this was a recent acquisition of *M. hominis* (IgM response), but also long-lasting antibody immunity (IgG). Did the patient have a previous history of *M. hominis* infection? Please also include a statement regarding ethics and the use of the serum from this patient.

The patient had a true *M. hominis* infection. Indeed, *M. hominis* was responsible for a post-partum endometritis. *M. hominis* was isolated by culture from the patient's placenta and also from the peripheral samples of her newborn. Remnant serum from the mother was anonymously collected two days after childbirth in the context of routine clinical care, explaining the presence of an IgM response in the Western blot. There was no information about a previous history of *M. hominis* infection.

As suggested, we specified the clinical context of the serum collection, lines 192-194 of the revised manuscript.

Informed consent is not required in France for the use of remnants of microbiological samples in the quality assurance of diagnostic methods according to the national legislation. The collection and use of anonymized samples were approved by the French Personal Data Protection Authority (CNIL, n°10.362). As suggested, the statement regarding ethics for the use of the patient serum was provided in the “Informed Consent Statement” section, lines 648-652 of the revised manuscript.

Line 183: What was the reason for the albumin depletion?

Serum was depleted of albumin because albumin has a molecular weight of 68 kDa, which is close to that of the IgG heavy chain (55 kDa intact, 44 cut) and IgM heavy chain (70 kDa, 59 cut). Albumin is highly abundant in serum, and consequently, is the largest protein band visible when performing a SDS-PAGE on a serum sample. As a result, the large albumin band has a tendency to deform and mask the much less abundant IgG/IgM bands (Nottelet *et al.*, Sci Adv, 2021). We therefore preferred to lower the albumin concentration through depletion in order to have a cleaner IgG/IgM band.

As suggested, we added these clarifications and the reference 27 (Nottelet *et al.*, Sci Adv, 2021) in the material and method section, lines 194-197 of the revised manuscript.

Line 187: 1.108 do you mean 1×10^8

You're right, we modified this point in the revised manuscript, line 200, as suggested.

Line 330: This sentence seems to be partial 'efficient M.'

Sorry, the sentence was incomplete. The complete sentence "However, the chosen 212 bp putative regulatory region including the main sequence determinants (Figure S1) was not suitable for efficient *M. hominis* transformation" was specified in the revised manuscript, line 342-343.

Line 341 & 342: Please give the full genus name for bacteria on the first use

As suggested, we mentioned the full genus name of bacteria in the discussion, lines 357-358 of the revised manuscript.

Reviewer comments:

Reviewer #1 (Comments for the Author):

The authors have answered the comments satisfactorily, so that in the manuscript the few passages that were difficult to understand have disappeared.

They have improved the traceability of their data.

Thank you for these comments.

However, in supplementary table 2, column K, the accession number of the homologs refers to the genome sequence of the isolate, the homolog protein given in column J derived from. It would be easy to specify the accession number of each protein (e.g. in replacing "MHO_0350 in NC_013511.1" by WP_012855330.1) In my mind, easy accessibility of data will facilitate scientific exchange and discussion.

As suggested, in supplementary table 2, the accession number of each protein was specified in the column K instead of the accession number of the genome sequence of the isolate.

If still possible the authors should use a non-proportional font in the multiple sequence alignment of Fig. S1 D.

As suggested, in supplementary figure 1, we provided an alternative representation using a non-proportional font in the multiple sequence alignment of Fig. S1 D.

Reviewer #2 (Comments for the Author):

Thank you for addressing the points raised. This is very interesting work and I look forward to seeing it published.

Thank you for these comments.

July 18, 2023

Prof. Sabine Pereyre
Univ. Bordeaux, CNRS
UMR 5234 Fundamental Microbiology and Pathogenicity
146 rue Léo Saignat
Bordeaux 33076
France

Re: Spectrum01873-23R2 (Improved transformation efficiency in *Mycoplasma hominis* enables disruption of the MIB-MIP system targeting human immunoglobulins)

Dear Prof. Sabine Pereyre:

Congratulations!

Your manuscript has been accepted, and I am forwarding it to the ASM Journals Department for publication. You will be notified when your proofs are ready to be viewed.

Sincerely,

Olivier Neyrolles, PhD
Editor, Microbiology Spectrum
